# A mitochondrial megachannel resides in monomeric $F_1F_O$ ATP synthase

Nelli Mnatsakanyan [1]*, Marc C. Llaguno [2], Youshan Yang[3], Yangyang Yan[3], Joachim Weber[4], Fred J. Sigworth [3] & Elizabeth A. Jonas [1]*

Purified mitochondrial ATP synthase has been shown to form $Ca^{2+}$-activated, large conductance channel activity similar to that of mitochondrial megachannel (MMC) or mitochondrial permeability transition pore (mPTP) but the oligomeric state required for channel formation is being debated. We reconstitute purified monomeric ATP synthase from porcine heart mitochondria into small unilamellar vesicles (SUVs) with the lipid composition of mitochondrial inner membrane and analyze its oligomeric state by electron cryomicroscopy. The cryo-EM density map reveals the presence of a single ATP synthase monomer with no density seen for a second molecule tilted at an 86° angle relative to the first. We show that this preparation of SUV-reconstituted ATP synthase monomers, when fused into giant unilamellar vesicles (GUVs), forms voltage-gated and $Ca^{2+}$-activated channels with the key features of mPTP. Based on our findings we conclude that the ATP synthase monomer is sufficient, and dimer formation is not required, for mPTP activity.

[1] Section of Endocrinology, Department of Internal Medicine, Yale University, New Haven, CT, USA. [2] Center for Cellular and Molecular Imaging, Yale University, New Haven, CT, USA. [3] Department of Cellular and Molecular Physiology, Yale University, New Haven, CT, USA. [4] Department of Chemistry and Biochemistry, Texas Tech University, Lubbock, TX, USA. *email: nelli.mnatsakanyan@yale.edu; elizabeth.jonas@yale.edu

M itochondrial $F_1F_O$ ATP synthase catalyzes the synthesis of ATP from ADP and inorganic phosphate during the final step of oxidative phosphorylation. While the main function of ATP synthase is to synthesize ATP, it can also run in reverse to hydrolyze ATP and energize the inner mitochondrial membrane under certain pathological and physiological conditions[1]. ATP synthase is a rotational molecular machine which has very complex structure: it consists of two subcomplexes, the hydrophilic $F_1$, which contains the catalytic nucleotide binding sites, and the membrane-embedded $F_O$[2–4]. ATP synthase uses the electrochemical transmembrane gradient of protons to drive the rotation of $F_O$ and initiate ATP synthesis in $F_1$ catalytic centers. Tight coupling between $F_O$ and $F_1$ subcomplexes is required for highly efficient rotational catalysis.

Despite its role as the main energy-producing enzyme, ATP synthase has been reported recently to also play a role in the mitochondrial inner membrane depolarization caused by the channel activity of the mitochondrial permeability transition pore (mPTP)[5–8]. The mitochondrial permeability transition (mPT) was first found to be correlated with mitochondrial inner membrane channel activity several decades ago: patch-clamp recordings of mitoplasts showed that the inner membrane contains a nonselective, high-conductance megachannel, which is sensitive to calcium, adenine nucleotides, and suspected to be the mPT pore[9–14]. In addition, patch-clamp recordings in neuronal pre-synaptic terminals revealed that prolonged activation of mitochondrial channel activity occurs during the calcium influx that is required for synaptic transmission[15]. Despite extensive research in the field, the exact molecular composition of the mPTP is still mysterious. mPTP was suggested previously to form from different mitochondrial inner membrane proteins including the adenine nucleotide translocator (ANT), the voltage-dependent anion channel (VDAC) and the phosphate carrier (PiC)[16–19]. However, the genetic ablation of these proteins revealed that they constitute regulatory components rather than the pore of the mPT[20–22]. Furthermore, the depletion of these proteins only desensitizes mPTP to $Ca^{2+}$ overload but does not eliminate the cyclosporin A (CsA)-sensitive mPT[20–22]. CsA inhibits the mPTP by interacting with its well-known regulator, a matrix protein cyclophilin D (CyPD), which was recently shown to bind to the oligomycin sensitivity-conferring protein (OSCP) subunit of the ATP synthase[23]. The interaction of CyPD with OSCP was suggested to trigger the opening of a large conductance channel found in ATP synthase upon high matrix $Ca^{2+}$ overload[5–7]. A recent report, however, suggested that complete knockout of the main membrane-embedded component of the ATP synthase, the c-subunit, resulted in no change in the sensitivity of mPT to calcium[24,25]. In contrast, patch-clamp recordings of c-subunit knockout mitoplasts were found to lack MMC activity[26]. Interestingly, other reports clearly show that partial depletion or knockdown of c-subunit attenuates mPT[6,27,28]. Another possible candidate for forming a channel within the ATP synthase was suggested to be the location between the ATP synthase dimers[5,7,29]. Subunits e and g were suggested to mediate the high-conductance channel formation at the dimer interface in yeast ATP synthase[29,30].

Mitochondrial $F_1F_O$ ATP synthase is known to form V-shaped dimers from two identical monomers in mitochondrial inner membranes[31,32]. The formation of ATP synthase dimers was shown to be essential for biogenesis and curvature of the inner membrane cristae[33,34]. Recent reports have shown that purified and detergent-solubilized dimers of $F_1F_O$ ATP synthase, when incorporated into lipid bilayers, form $Ca^{2+}$-activated channels, with the key features of the megachannel associated with the mPT[5,8]. However, the question still remains open whether the ATP synthase dimer or the monomer alone can form a channel.

In this work we aimed to identify the minimal unit or oligomeric state of ATP synthase required for the formation of an ion channel.

We purify monomeric $F_1F_O$ ATP synthase from porcine heart mitochondria and reconstitute it into small unilamellar vesicles (SUVs). Single-particle cryo-EM analysis of reconstituted ATP synthase confirmed the monomeric state of ATP synthase after reconstitution. We show that fully assembled and catalytically active ATP synthase monomers incorporated into GUVs form a high-conductance voltage-gated and $Ca^{2+}$-activated channel, sensitive to ATP, the ATP synthase inhibitor oligomycin, and other known mPTP modulators.

## Results

**ATP synthase forms monomers after reconstitution.** Dodecyl maltoside (DDM)-solubilized ATP synthase was purified from porcine heart mitochondrial membranes by large-scale purification. The monomeric state of the purified protein was confirmed by FPLC and blue-native PAGE (Fig. 1a, b, sample 2). In contrast to DDM, solubilization of mitochondrial membranes with the mild detergent digitonin led to the formation of both monomeric and dimeric ATP synthase (Fig. 1b, sample 1).

DDM-solubilized ATP synthase was reconstituted into SUVs (size range ~25–50 nm) by prolonged dialysis. ATP and $Mg^{2+}$ were included in all the buffers used for protein purification and reconstitution to ensure the stability of the purified protein complex. The lipids used for reconstitution (phosphatidylethanolamine (PE), phosphatidylcholine (PC), and cardiolipin (CL), 35%:35%:30%) closely mimicked the lipid composition of the mitochondrial inner membrane[35].

The presence of various ATP synthase subunits after purification and reconstitution was confirmed by liquid chromatography-mass spectrometry (LC-MS/MS) and visualized by SDS-PAGE (Fig. 1c, Supplementary Data 1). According to our immunoblot analysis the ATP synthase preparation was free of contamination with adenine nucleotide transporter 1 (ANT1), which is the main ANT isoform in heart mitochondria[36] (Supplementary Fig. 1a). CypD was also missing from the protein preparation consistent with previous reports suggesting its loose association with ATP synthase[37] (Supplementary Fig. 1a). LC-MS/MS analysis of purified samples showed a residual contamination with ANT1 (0.54% relative to ATP synthase β subunit abundance) but not with CypD, VDAC, or PiC. The purified sample demonstrated oligomycin-sensitive ATP hydrolysis activity before and after reconstitution into liposomes, which confirmed that the protein is in its fully assembled and coupled conformation (Fig. 1d).

Single-particle cryo-EM analysis of ATP synthase-reconstituted vesicles was performed to evaluate the oligomeric state of ATP synthase after reconstitution. SUVs reconstituted with functionally active ATP synthase were frozen on C–flat grids covered with a thin carbon layer (5–8 nm) (Fig. 1e, f). We found that the insertion of ATP synthase into liposomes was unidirectional, with the hydrophilic $F_1$ head pointing out of the vesicle (Fig. 1e). This was confirmed by the immunoblot analysis of sodium bromide treated liposomes. Sodium bromide removes $F_1$ if it is located outside but not inside the liposomes (Supplementary Fig. 1b).

A 96-pixel box size (2.8 Å/pixel) was used for particle picking after vesicle subtraction (Fig. 2a). 7795 particles were used for reconstruction of a 3D density map, which had ~19 Å resolution (Figs. 2b, c, S1c). The cryo-EM structure of bovine ATP synthase (5ARA)[4] was used for fitting into the 3D density map of our porcine ATP synthase structure (Fig. 2d).

It was recently reported in yeast and metazoans that the mitochondrial ATP synthase forms dimers at a specific 86° angle between the monomers[38,39]. To test whether ATP synthase

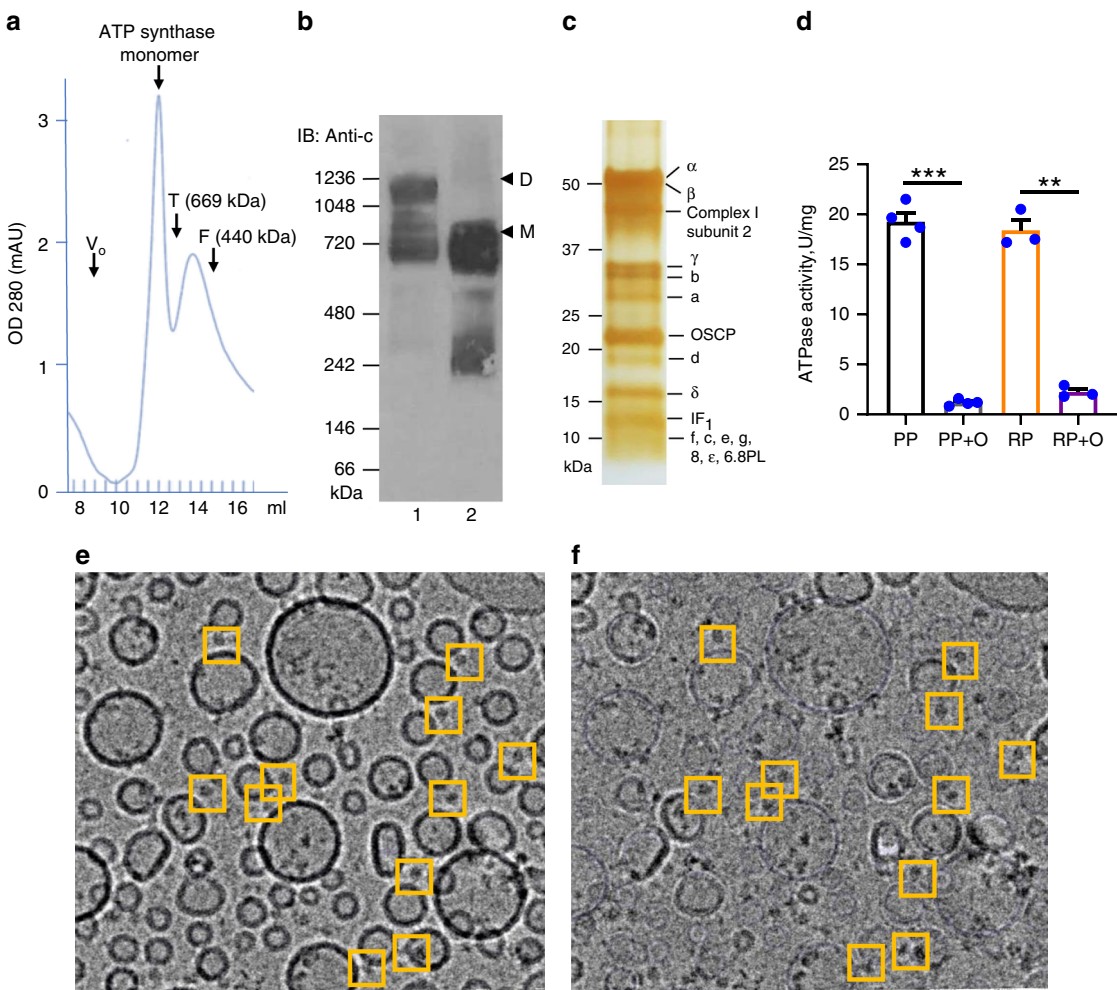

**Fig. 1 DDM-purified ATP synthase forms monomers. a** Size exclusion chromatography (SEC) profile of purified ATP synthase monomer. A representative example of three independent runs is shown. Void volume of SEC column ($V_o$) and elution volume of protein molecular weight markers, thyroglobulin (T, 669 kDa) and ferritin (F, 440 kDa) are shown. **b** Blue-native PAGE and immunoblot analysis of porcine heart mitochondria solubilized with digitonin (sample 1) and porcine ATP synthase purified from DDM-solubilized mitochondria (sample 2). ATP synthase c-subunit antibody was used to detect monomeric (M) and dimeric (D) ATP synthase. Molecular weight markers are indicated on the left side of the gel. Representative blot of three independent experiments is shown. **c** SDS-PAGE analysis of DDM-purified monomeric ATP synthase after reconstitution in small unilamellar vesicles. Silver stain was used for protein band visualization. Molecular weight markers are indicated on the left side of the gel. All bands are identified based on confirmation either by LC/MS/MS or immunoblot analysis (Supplementary Fig. 1d, Supplementary Data 1). **d** ATP hydrolysis activity of purified ATP synthase, per mg protein, before and after reconstitution into lipid vesicles. PP and RP refer to purified protein and reconstituted protein, respectively; O refers to oligomycin. Oligomycin-sensitive ATP hydrolysis activity was measured to assess the coupling of purified protein ($n = 4$ for PP and $n = 3$ for RP, ***$P < 0.05$, **$P < 0.05$, paired $t$ test was used, error bars refer to SEM). **e** Micrograph showing the cryo-EM images of monomeric ATP synthase reconstituted into small unilamellar vesicles before and **f** after vesicle subtraction. Each yellow box (125 × 125 Å) indicates one ATP synthase particle pointing out of the lipid vesicle. Vesicles were frozen on C-flat grids covered with a thin carbon layer (5–8 nm). The source data underlying **b**, **c**, **d** are provided as a Source Data file.

dimers were also included in our cryo-EM analysis we used a larger box (128-pixel box, 4.2 Å/pixel) for particle extraction, chosen to accommodate any ATP synthase dimers (Fig. 2e). 6655 particles were used for the reconstruction of the second 3D map after vesicle subtraction, which had 30 Å resolution (Fig. 2f–h). The second map had lower resolution than the first one, since a larger box was used without any mask for 3D reconstruction. The first 19 Å reconstruction was used as a reference for the second map, and density for the second ATP synthase in a dimer, at a specific 86° angle, was not found in the map, although some monomers were seen to be closely spaced. In Fig. 2f we demonstrate the possible location of a second monomer at the 86° angle position if it were to be present in the map.

These results are consistent with the recent cryo-electron tomography study of purified reconstituted ATP synthase from

*Polytomella* sp. and *Y. lipolytica*[40]. This report showed that digitonin-solubilized dimers of ATP synthase assemble spontaneously into dimeric rows upon reconstitution into liposomes and induce membrane curvature, while DDM-solubilized ATP synthase forms only monomers, which are randomly distributed in liposomes, without forming rows or curving the membrane[40].

**ATP synthase forms a voltage-gated channel sensitive to Ca²⁺.**
We performed patch-clamp recordings to characterize the channel activity formed by the monomeric ATP synthase. We fused the ATP synthase reconstituted SUVs with GUVs in the presence of SM-2 Bio-Beads to fully remove the residual amount of detergent if it were to be still present in the SUV sample. GUVs were then used for patch-clamp recordings (Fig. 3a, b). The single

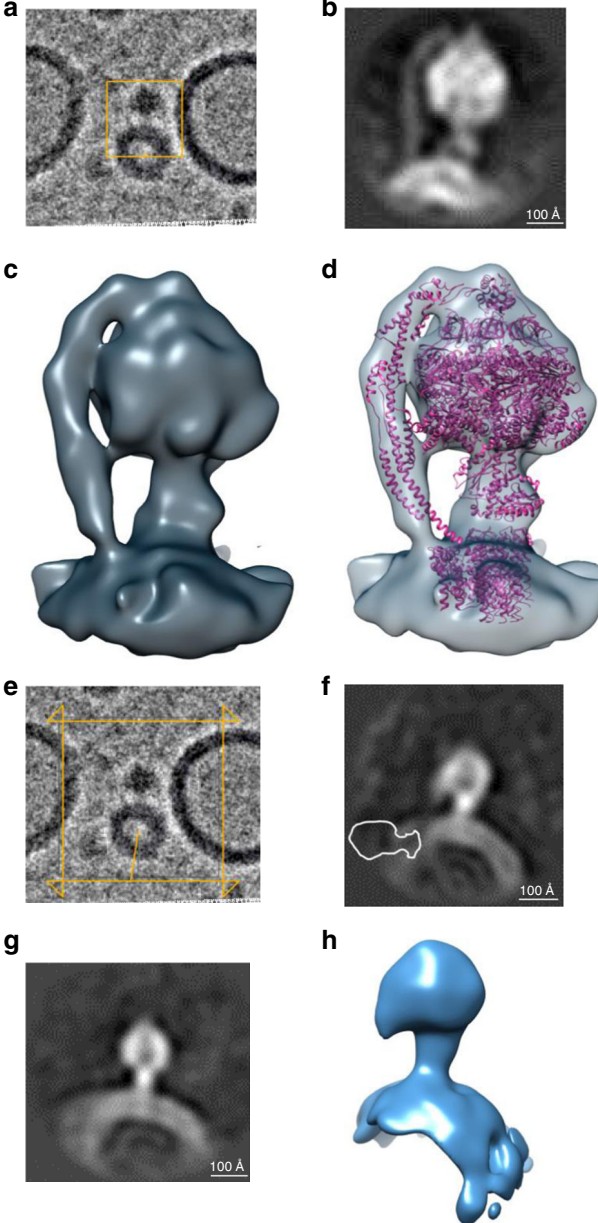

**Fig. 2 ATP synthase forms monomers after reconstitution. a** Cryo-EM image of small unilamellar vesicles reconstituted with ATP synthase. For initial particle picking a 96-pixel box at 2.8 Å/pixel was used for particle picking and 3D reconstruction. **b** The resulting 19 Å resolution map of porcine ATP synthase monomer in liposomes. 7795 particles were used for reconstruction of the 3D map after vesicle subtraction. **c** 19 Å cryo-EM structure of ATP synthase in a liposome. **d** The cryo-EM structure of bovine ATP synthase (5ARA) (4) was fitted into the 3D reconstruction of porcine ATP synthase. **e** The cryo-EM image shown in **a** of small unilamellar vesicle-reconstituted ATP synthase is shown here again with a larger box size (128-pixel box, 4.2 Å/pixel), which was used to accommodate dimers. **f, g** The side and front slice views of the resulting 30 Å resolution map from 6655 particles. Density for a second ATP synthase monomer was not found in the map. The outlined image in **f** represents the expected location of the second monomer (at 86° angle position) if it were to be present in the map. **h** A surface rendered view of the 30 Å resolution map. **c**, **d**, and **h** of this figure were generated with Chimera[62]. ATP synthase models from **a** and **e** have been deposited into the Electron Microscopy Data Bank (accession codes EMD-21001, EMD-21002, respectively).

channel recording in Fig. 3d demonstrates that the reconstituted ATP synthase has large conductance single channel activity. The amplitude histogram reveals that the peak open state of this channel has a conductance of 1.8 nS (Fig. 3e). The mean peak conductance of the channel activity in a total of 30 patches was ~1.3 nS (Fig. 3c). The same peak conductance was observed regardless of the lipid composition used (PC only or PE:PC:CL, Fig. 3c compared with Supplementary Fig. 3d). Figure 3f demonstrates the voltage dependence of channel activity, which is inwardly-rectifying. These data suggest that voltage change alone is sufficient for triggering channel activity. The empty liposomes did not exhibit channel activity at any voltage (Fig. 3c, f). In several patches we also observed the channel activity in a multi-conductance mode, with multiple open states (Supplementary Fig. 2a, b). These different states do not all share the same amplitude, indicating that these states most likely represent subconductance modes of a single channel with maximum conductance of about 1 nS, rather than multiple single channels (Supplementary Fig. 2a).

One of the biophysical characteristics of the mPTP is its sensitivity to adenine nucleotides[9,12,14]. Figure 3g shows a patch-clamp recording of channel activity before and after ATP addition. Inhibition of channel activity is observed upon addition of 1 mM ATP to the bath (Fig. 3g, h). The ramp voltage recording shows that ATP inhibits the channel activity at all voltages (Fig. 3i).

The mPTP is also known to be sensitive to calcium, which opens the channel[13,15]. Therefore, we set out to determine the effect of calcium on ATP synthase channel activity. Figure 4a shows a continuous recording of a channel in the presence of increasing concentrations of $Ca^{2+}$. $Ca^{2+}$ was added twice during the recording in 500 μM increments. The addition of 1 mM $Ca^{2+}$ increased the frequency of channel openings ~3 times (Fig. 4a, d), while ATP, added at the end of the experiment, inhibited the channel activity in the presence of $Ca^{2+}$ (Fig. 4a, c). In order to determine the peak amplitude of conductance of the channel in the presence of calcium, we preexposed ATP synthase-reconstituted liposomes to 2 mM $Ca^{2+}$ before forming a patch (Fig. 4b). We observed large conductance activity with a peak conductance of ~1.3 nS with inhibition of the channel activity upon ATP addition (Fig. 4b, c). Therefore, we found that $Ca^{2+}$ increased the frequency but not the peak conductance of the channel activity.

The ramp voltage recording of channel activity shows that channel activity often becomes non-rectifying in the presence of $Ca^{2+}$: $Ca^{2+}$ enhances the channel activity at positive voltages (Fig. 4e). ATP inhibits the channel activity at all voltages in the presence of $Ca^{2+}$ (Fig. 4e).

**Pharmacological modulation of ATP synthase channel activity.** As previously defined in studies of the mPTP, $Ba^{2+}$ and other divalent cations are known to antagonize the effects of $Ca^{2+}$ or to close the channel independently of $Ca^{2+}$[13]. We herein find that the channel formed by the ATP synthase monomer closes immediately upon addition of 1 mM $Ba^{2+}$ (Fig. 5a, d). The channel stayed closed in the presence of $Ba^{2+}$ throughout the recording unless the $Ba^{2+}$ was washed out (Supplementary Fig. 3a). This inhibition was reversible, since the channel reopened after washing the patch with the recording buffer (Supplementary Fig. 3a) but closed again immediately after subsequent addition of 1 mM $Ba^{2+}$ (Supplementary Fig. 3a). We treated ATP synthase reconstituted liposomes with sodium bromide (3 M), to strip $F_1$ (Supplementary Fig. 1b). 4 mM $Ba^{2+}$ added in two subsequent additions during continuous recording failed to close the channel in $F_1$-stripped liposomes (Supplementary Fig. 3b). These data suggest that the effect of $Ba^{2+}$ is

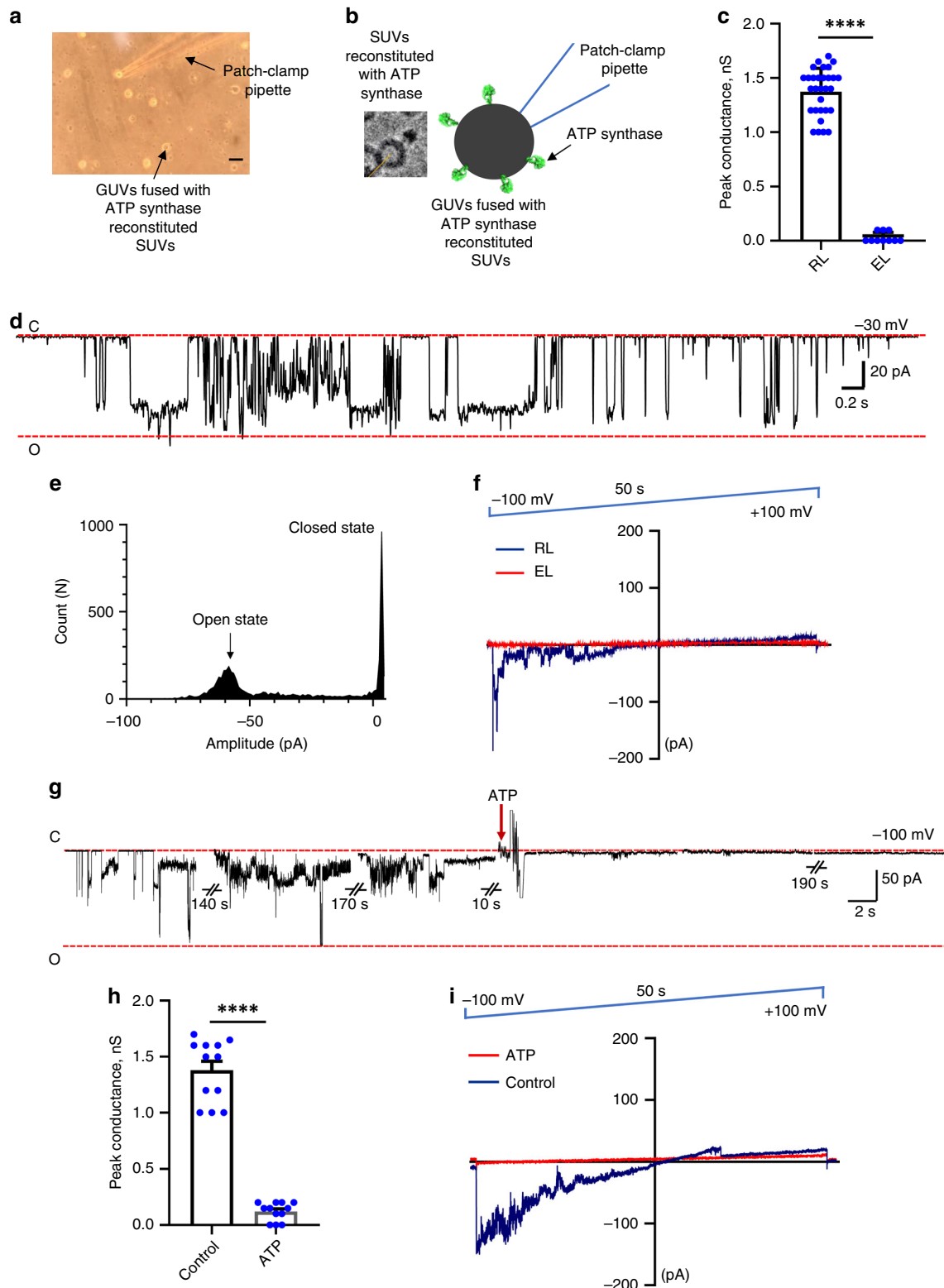

reversible and occurs in an $F_1$-dependent manner, showing that the binding site for $Ba^{2+}$ is located in $F_1$. Sensitivity to $Ba^{2+}$ in the recordings was indistinguishable between GUVs prepared with two different lipid compositions (PC only or PE:PC:CL, Fig. 5a, d and Supplementary Fig. 3c, d, respectively).

$Gd^{3+}$, another cation known to inhibit mPTP, was also tested on the channel activity formed by ATP synthase. A representative trace (Fig. 5b) and group data (Fig. 5d) show immediate inhibition of the channel upon 1 mM $Gd^{3+}$ addition.

Oligomycin A is a specific inhibitor of ATP synthase with a known binding site within the ATP synthase c-subunit ring[41]. It is proposed that binding of oligomycin inhibits ATPase activity by causing a conformational change in the $F_O$ portion of the complex that is transmitted to $F_1$ through OSCP[42].

**Fig. 3 ATP synthase forms a large voltage-gated channel. a** Photomicrographic image during a patch-clamp recording showing the patch-clamp pipette sealed onto a liposome. Scale bar, 50 μm. **b** Left image, the cryo-EM image of monomeric ATP synthase reconstituted into an SUV and Right image, diagram of a patch-clamp pipette sealed onto the ATP synthase reconstituted giant unilamellar vesicles. **c** Group data of the peak conductance of channel activity of empty and ATP synthase reconstituted liposomes ($n = 10$ for empty, EL, and $n = 30$ for ATP synthase reconstituted liposomes, RL, ****$P < 0.0001$, unpaired $t$ test was used). Only ATP-sensitive patches were included in the statistical analysis performed before and after ATP. **d** Representative patch-clamp recording of $F_1F_O$ ATP synthase reconstituted liposomes. The channel shows predominantly single conductance mode. C, indicates the closed state, O indicates the open state of the channel. **e** Amplitude histogram of the same current trace showing the main open state of the channel with 1.8 nS conductance. **f** Representative ramp voltage recording of an empty (EL, red trace) or an ATP synthase reconstituted liposome (RL, blue trace). **g** Representative patch-clamp recording of ATP synthase channel activity before and after addition of ATP, showing inhibition of channel activity upon ATP (disodium salt) addition. The channel remained closed after addition of ATP, or had very low conductance, infrequent activity until the end of the recording. **h** Group data and **i** current-voltage relationship of ATP synthase channel activity in the presence and absence of 1 mM ATP. $n = 12$, ****$P < 0.0001$, paired $t$ test was used, error bars refer to SEM. The source data underlying **c** and **h** are provided as a Source Data file.

Figure 5c, d show that the addition of oligomycin immediately closes the channel. This result strongly suggests that ATP synthase itself and not any other contaminants, i.e., oligomycin-insensitive proteins, are causing the channel activity. This allows us to suggest that the oligomycin inhibition induces conformational changes in ATP synthase structure, which locks the channel in its closed conformation. The oligomycin-inhibited structure of ATP synthase will reveal more insights about this conformation. Interestingly, it was shown that an oligomycin A derivative (1,3,8-triazaspiro[4.5]decane) inhibits the opening of mPTP by targeting directly the c-subunit of ATP synthase and has beneficial effects in a model of myocardial infarction[43].

Based on our findings we conclude that purified and functionally active ATP synthase monomer is sufficient to form a high-conductance voltage-gated channel with biophysical characteristics of the mPTP.

## Discussion

The formation of ATP synthase dimers is vital for the biogenesis of mitochondrial cristae, which are heavily folded mitochondrial inner membrane structures designed to increase the surface for oxidative phosphorylation[31–34]. Cristae were suggested to work as proton traps, important for maintaining the electrochemical gradient of protons generated by the electron transport chain across the inner mitochondrial membrane[44]. This well-designed geometry of the mitochondrial inner membrane allows efficient ATP synthesis during aerobic respiration. The flow of protons through the ATP synthase proton translocator, located between the a and c-subunits, rotates the central stalk of ATP synthase and drives the synthesis of ATP in $F_1$ catalytic centers[45]. Perfect chemo-mechanical coupling was suggested to be required for efficient ATP synthesis in vitro[46]. However, uncoupling of these processes most likely occurs during certain physiological and pathological conditions in vivo due to the physiological and pathological opening of the mPTP. Despite its clinical importance, the molecular identity of such a mitochondrial leak channel has been enigmatic.

In this study we combined single-particle cryo-EM analysis with electrophysiological techniques to characterize channel activity formed by the ATP synthase. The channel forming activity of purified and reconstituted ATP synthase has been reported previously[5,6,29,30]. It was suggested that the mPTP is formed at the membrane interface between two adjacent $F_O$ sectors of ATP synthase and that ATP synthase dimers are necessary for channel formation[5,29,30,47]. Highly purified ATP synthase tetramers and dimers, but not monomers, were shown to form channels in planar lipid bilayer experiments in these reports. However, the oligomeric state of ATP synthase after reconstitution was not assessed before the sample was tested for its channel activity. The authors examined electrophysiological behavior of samples cut out from BN-PAGE. According to[47], the gel eluted samples were not characterized biochemically before they were used in electrophysiological recordings, therefore the oligomeric state of samples during the recordings remains unknown.

Here we performed single-particle cryo-EM analysis of ATP synthase after reconstitution into SUVs, which revealed the presence of ATP synthase monomers in the density map. We demonstrate that the same functionally active ATP synthase monomers in SUVs form voltage-gated channels when fused into GUVs. We find that the channel activity can be enhanced by $Ca^{2+}$ and inhibited by ATP. We also find that $Ba^{2+}$ and $Gd^{3+}$ inhibit channel activity, consistent with previously known properties of the mPTP.

CypD is a well-known regulator of mPTP, which can trigger channel opening and sensitize the mPTP to $Ca^{2+}$[23,48]. The immunosuppressant drug CsA inhibits CypD-induced channel openings. Even though CypD was shown to interact with the ATP synthase subunit OSCP[37,49], we did not detect it in our purified sample. The absence of CypD in the ATP synthase preparation precluded the possibility of analyzing its effects during our studies. It is possible that this interaction occurs under specific conditions or that it is simply not preserved during the multistep ATP synthase purification procedure.

OSCP has been reported to be the site of interaction not only for CypD, but also for other endogenous and pharmacological modulators of mPTP[50–52]. The role of OSCP is crucial in coupling ATP synthase $F_O$ with $F_1$ and it seems to be an important site of modulation of ATP synthase leak channel activity[49,53,54].

We observe the oligomycin-sensitive ATP hydrolysis activity by ATP synthase before and after reconstitution into liposomes (Fig. 1d). We also show that oligomycin inhibits the ATP synthase channel conductance (Fig. 5c, d). This allows us to suggest that oligomycin inhibition induces conformational changes in the $F_1$:$F_O$ interaction, leading to the closed channel conformation of ATP synthase. Future structural analysis of ATP synthase inhibited by oligomycin will perhaps reveal this conformation. Surprisingly, a recent structure of ATP synthase tetramers inhibited with $IF_1$ revealed an unexpected structure of the $F_O$ region[55]. The central lumen of $c_8$-rings was shown to be filled, not by phospholipid plugs, as it has been reported previously[56], but instead by a 40 amino acid-long alpha-helical segment. The authors assigned ATP synthase membrane subunit 6.8PL to this density, although more studies are needed to definitively establish the identity of this protein. According to this structure, the C-termini of the four e subunits are bent toward the four $c_8$-rings in each protomer and interact with the C-terminal ends of the 6.8PL helices, which fill the central hole inside the $c_8$-rings. We speculate that this interaction between subunit e and 6.8PL may be important for removing 6.8PL from the c-subunit lumen and thereby inducing channel activity.

We have previously shown that the purified and reconstituted c-subunit ring forms a channel with similar characteristics to

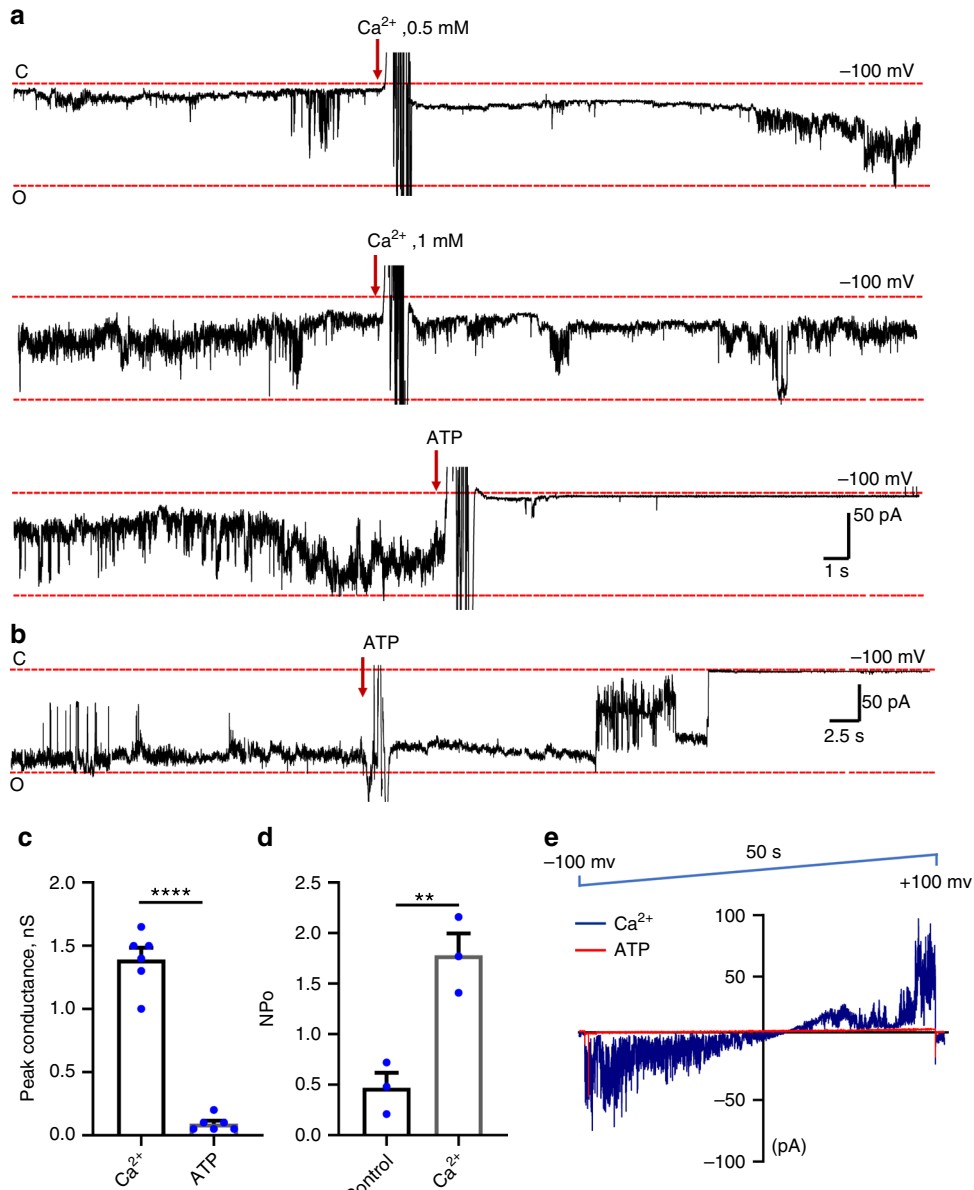

**Fig. 4 ATP synthase forms a channel sensitive to $Ca^{2+}$. a** A single continuous recording of ATP synthase reconstituted liposomes in the presence of increasing concentrations of $Ca^{2+}$. $Ca^{2+}$ was added twice during the recording, yielding final concentrations of 500 μM and 1 mM. ATP still inhibits the channel activity in the presence of $Ca^{2+}$. The current trace is representative of 6 recordings. **b** A single continuous recording of ATP synthase channel in the presence of 2 mM $Ca^{2+}$. ATP synthase reconstituted liposomes were pre-exposed to 2 mM $Ca^{2+}$ before forming a patch. The current trace is representative of four recordings. **c** Group data for the peak conductance of ATP synthase channel activity in the presence of $Ca^{2+}$ with and without 1 mM ATP, $n = 6$, ****$P < 0.0001$, paired $t$ test was used. **d** Group data for the open probability of channels (NPo) in response to $Ca^{2+}$, $n = 3$, **$P < 0.0028$, paired $t$ test was used. Both subconductance and peak conductance levels were included in NPo measurements. **e** Current-voltage relationship of ATP synthase channel activity in the presence of $Ca^{2+}$ with and without 1 mM ATP. $n = 6$, ****$P < 0.0001$, error bars refer to SEM. The source data underlying (**c** and **d**) are provided as a Source Data file.

those of the ATP synthase monomers[6]. Nevertheless, the relationship between the c-subunit channel and mPTP channel has been recently questioned by a study that used c-subunit knockout HAP1-A12 cells[25]. That study suggested that mitochondria can still undergo $Ca^{2+}$-induced and CsA-sensitive membrane depolarization in the absence of the c-subunit[25]. Although the c-subunit may not be the only inner mitochondrial membrane protein required for mPTP formation, it is so far the main and largest conductance contributor to inner membrane depolarization under conditions of $Ca^{2+}$ overload and cell stress[6]. This has been supported by recent patch-clamp analysis[26] of the same HAP1-A12 c-subunit knockout cells used in[25]. These patch-

clamp recordings clearly demonstrated the lack of mPTP channel activity in c-subunit knockout cells. However, channels with significantly lower conductance but with sensitivity to CsA and to the ANT inhibitor bongkrekic acid were detected during these recordings[26]. Many other low conductance channels of the inner mitochondrial membrane with such characteristics could also contribute to membrane depolarization upon $Ca^{2+}$ overload.

In this study we show that purified and functionally active ATP synthase monomers of porcine heart mitochondria form voltage-gated and $Ca^{2+}$-activated channel activity with the known biophysical characteristics of mPTP. These findings present strong evidence that ATP synthase monomer is sufficient to form a

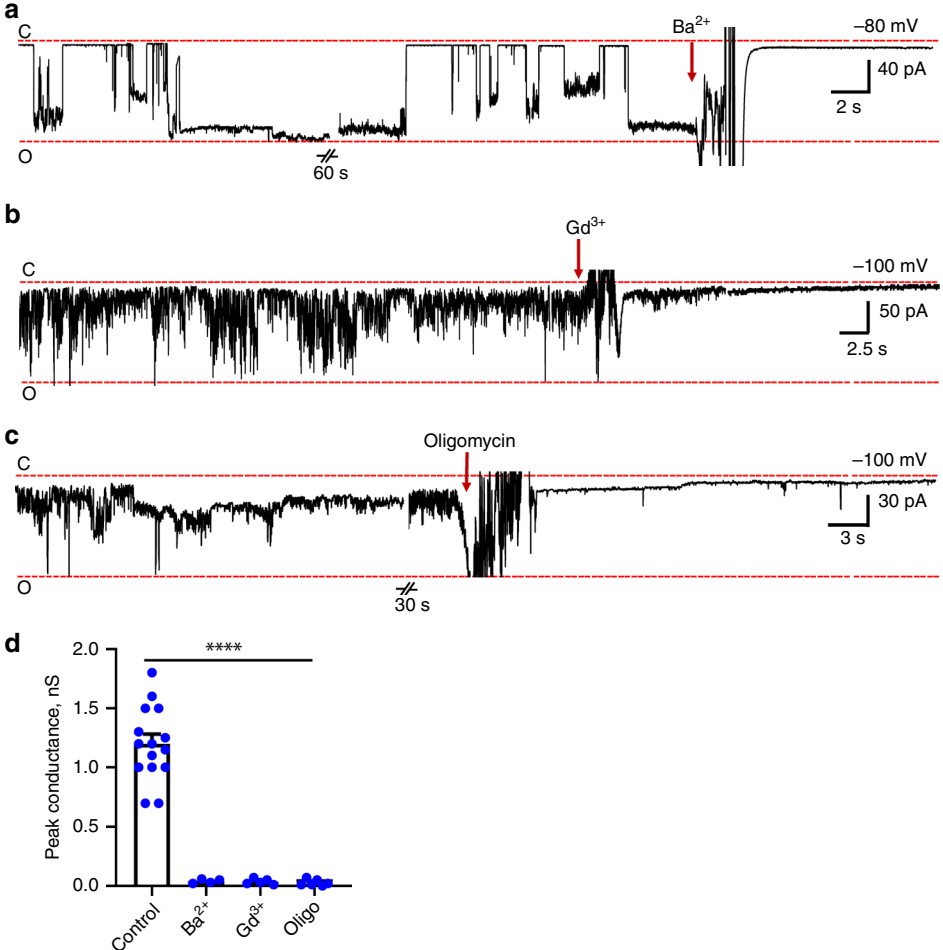

**Fig. 5 Pharmacological modulation of ATP synthase channel activity.** Representative patch-clamp recording of $F_1F_O$ ATP synthase reconstituted liposomes showing sensitivity to **a** $Ba^{2+}$ (1 mM), $n = 4$, **b** $Gd^{3+}$ (1 mM), $n = 5$, and **c** oligomycin A (5 μg/ml), $n = 6$. During the oligomycin experiment, the closure of the channel was not observed when the vehicle, ethanol, was added during the recordings. C, indicates the closed state, O indicates the open state of the channel. **d** Group data for the peak conductance of ATP synthase channel activity for the conditions noted in **a**, **b**, and **c**. One-way ANOVA test was used for statistical analysis, ****$P < 0.0001$, error bars refer to SEM. The source data underlying **d** are provided as a Source Data file.

channel. This does not mean that the channel cannot form when the ATP synthase is in its dimeric state. However, the ATP synthase monomer is the minimal unit required for channel formation. In-depth structural analysis will reveal the open channel conformation of ATP synthase and will provide mechanistic insights into the conformational changes of ATP synthase required to open this pore of mPT.

## Methods

**Isolation of mitochondria and purification of ATP synthase**. Fresh porcine hearts were obtained immediately after slaughter and sub-mitochondrial vesicles (SMVs) were isolated as previously described[57]. Briefly, the heart was finely minced in Buffer 1 [225 mM mannitol, 70 mM sucrose, 1 mM EGTA, and 20 mM Tris (pH 7.2)], and the tissue was then homogenized with an Elvehjem potter. The homogenate was centrifuged at $1000 \times g$ for 4 min. The supernatant was transferred into a fresh tube, and the sediment was homogenized for a second time to access intermyofibrillar mitochondria. After centrifugation at $1000 \times g$, the supernatants were centrifuged at $9000 \times g$ for 10 min to sediment the mitochondria. The mitochondria were resuspended in Buffer 2 [225 mM mannitol, 70 mM sucrose, and 20 mM Tris (pH 7.2)] and centrifuged again at $9000 \times g$ for 10 min. The final mitochondrial pellet was suspended in 0.05–0.4 mL of Buffer 2. Isolated mitochondria were used immediately for SMV isolation or stored at −80 °C until further use. SMVs were solubilized on ice by using DDM (1 g/g protein) orCH-APSO and cholate to final concentrations of 1.6% (w/v) and 0.5% (w/v), respectively. Then, the suspension was centrifuged at $100,000 \times g$ for 1 h. Subsequently, 50% (w/w) PEG 6000 was added to the supernatant (final concentration, 7%). After 2 h incubation on ice the protein precipitate was removed by centrifugation at $50,000 \times g$ for 1 h and PEG 6000 was subsequently added to the supernatant (final

concentration, 2%). The protein precipitate was collected after incubation for 2 h on ice and centrifuged at $50,000 \times g$ for 1 h. The pellet, which contains ATP synthase, was solubilized in Buffer A (50 mM Tris-HCl, pH 8.0, 100 mM NaCl, 2 mM $MgSO_4$, 1 mM ATP, 0.1% DDM, 10% glycerol (v/v)). The solubilized sample was concentrated to 0.5 ml (Amicon centrifugal filters with 100 kDa molecular mass cutoff) and loaded onto a Superose 6 Increase 10/300 gel filtration column (GE Healthcare, USA) equilibrated with Buffer B (Buffer A with 0.1 mg/ml bovine heart polar lipid extract, Avanti). Fractions eluted at a retention volume of 11–12 ml were collected and used for further analysis. Protein concentrations were determined with a BCA Protein Assay Kit (Thermo Scientific). The protein molecular weight markers, thyroglobulin (T, 669 kDa), ferritin (F, 440 kDa), and BSA (66 kDa) were used for column calibration. The monomeric state of the purified protein was confirmed by blue-native PAGE. Purified protein was used immediately, or it was flash frozen in aliquots for further use. Purified ATP synthase from five independent preparations was used for ion channel recordings.

**SDS-PAGE and BNP-PAGE and western blot analysis**. Samples for SDS-PAGE were denatured for 2 min at 90 °C in a sample lysis buffer (50 mM Tris-HCl, pH 7.0, 4% (w/v) SDS, 10% (v/v) glycerol, and 0.1 M 1,4-Dithiothreitol) and separated on 4–16% polyacrylamide gels (Bio-Rad Mini Protean) by using Tris-HCl buffer system. Gels were stained with the silver stain kit for protein visualization (Pierce, Thermo Scientific) or wet-transferred on polyvinylidene fluoride (PVDF) membrane for the immunodetection of proteins. The membrane was blocked in 5% nonfat milk in TBS (Tris-buffered saline) and then probed with different antibodies: for ATP synthase α-subunit (1:1000 dilution, ab14748, Abcam), e-subunit (1:1000 dilution, ab122241, Abcam), g-subunit (1:1000 dilution, ab126181, Abcam), c-subunit (1:1000 dilution, anti-ATP5G1/G2/G3 antibody, ab180149, Abcam), ANT1 (1:1000 dilution, 32484, Sabbiotech) and CypD (1:1000 dilution, TA302580, Origene).

Porcine heart mitochondrial or purified ATP synthase samples were used for the blue-native PAGE analysis to assess the oligomeric state of ATP synthase. Isolated mitochondria (20 µg, verified by BCA assay) were solubilized with DDM (1 g/g protein) or digitonin (2 g/g protein) for 20 min on ice and centrifuged at $20,000 \times g$ for 30 min to remove the mitochondrial debris. Supernatants containing Coomassie Brilliant Blue G-250 (dye/detergent ratio 1:4) were separated on Bis-Tris 3–12% native gels (Invitrogen NativePAGE) by using the Bis-Tris buffer system. After separation the protein complexes were wet-transferred on a PVDF membrane, which was blocked in 5% nonfat milk in TBS and then probed with antibodies.

**Reconstitution of ATP synthase into liposomes.** DDM-solubilized ATP synthase monomers were reconstituted into SUVs (size range ~25–50 nm). The following lipid composition, closely mimicking that of the inner mitochondrial membrane[35], was used for making SUVs: PE:PC:CL (35%:35%:30%, Avanti). Liposomes were destabilized by sonicating the lipid mixture (final concentration, 10 mM) in 200 µL of Buffer C (50 mM Tris-HCl, pH 8.0, 150 mM NaCl) with DM (decyl-maltoside, final concentration, 30 mM) for 30 min. DDM-solubilized ATP synthase was mixed with the lipid/DM mixture in 2:1 ratio and incubated for 2 h at 4 °C. The protein/lipid/detergent mixture was dialyzed for 4 days in Buffer D (50 mM Tris-HCl, pH 8.0, 150 mM NaCl, 2 mM $MgSO_4$, 1 mM ATP) at 4 °C by using dialysis tubing (25000 MWCO, Cole-Parmer). The dialyzed sample was centrifuged at $15,000 \times g$ for 20 min and the supernatant was used immediately for cryo-EM grid preparation. The pellet containing relatively larger size SUVs was resuspended in Buffer E (10 mM HEPES, pH 7.3, 120 mM KCl and 50 mM NaCl) and fused with GUV's for electrophysiological recordings. GUVs were prepared as described previously[6]. Briefly, 50 µl of 50 mg/ml chloroform solution of PC was dried under a nitrogen stream in a vacuum to form a thin lipid bilayer film on a glass surface. For a separate recording experiment (Supplementary Fig. 3c, d), GUVs were prepared with the exact lipid composition of SUVs used for cryo-EM studies (PE:PC:CL, 35%:35%:30%). Empty or ATP synthase reconstituted SUVs were fused with GUVs by overnight incubation in Buffer E at 4 °C. SM-2 Bio-Beads (Bio-rad) were added to this mixture to remove the residual amount of detergent if it was still present in our SUV sample. Bio-Beads were removed by brief centrifugation. $F_1$ regions were stripped by incubating GUVs with 3 M sodium bromide for 30 min[58]. Stripped GUVs were collected after centrifugation at $15,000 \times g$ for 20 min. The absence of $F_1$ was confirmed by immunoblot analysis (Supplementary Fig. 1b).

**ATPase activity measurements.** ATPase activities were assayed in a buffer containing 50 mM $Tris/H_2SO_4$, 10 mM ATP, and 4 mM $MgSO_4$, pH 8.0, at 37 °C. The reaction was started by the addition of 5–10 µg enzyme solubilized in detergent or after liposome reconstitution and stopped after 2 min by the addition of SDS (final concentration, 10%, w/v). The released $P_i$ was measured as described[59]. One unit of enzymatic activity (U) corresponds to 1 µmol of ATP hydrolyzed (equivalent to 1 µmol of $P_i$ produced) per min, per mg protein. The effect of oligomycin on ATP hydrolysis activity was determined by addition of the inhibitor in ethanol solution (5 µg/ml, w/v).

**Electrophysiology.** The patch-clamp recordings of ATP synthase reconstituted GUVs were performed by forming a giga-ohm seal in intracellular solution (10 mM Hepes, pH 7.3, 120 mM KCl, 8 mM NaCl, 0.5 mM EGTA,) using an Axopatch 200B amplifier (Axon Instruments) at room temperature (22–25 °C). Recording electrodes were pulled from borosilicate glass capillaries (WPI) with a final resistance in the range of ~50 MΩ. Signals were filtered at 5 kHz using the amplifier circuitry. ATP and calcium were added into the bath during the patch-clamp recordings in 1:10 ratio (v/v) without perfusion. pCLAMP-10 software was used for patch-clamp electrophysiology data acquisition and analysis (molecular devices). All current measurements were adjusted for the holding voltage assuming a linear current-voltage relationship: the resulting conductances are expressed in pS according to the equation $G = I/V$, where $G$ is conductance in pS, $V$ is the membrane holding voltage in mV, and $I$ is the peak membrane current in pA-baseline electrode leak current. Group data were quantified in terms of conductance. All population data were expressed as mean ± SEM.

**Single-particle Cryo-EM analysis of ATP synthase.** Thin carbon films (thickness 5–8 nm) were evaporated on freshly cleaved mica in a Leica ACE 600 evaporator. ATP synthase reconstituted liposomes were applied to the glow-discharged C-flat grids covered with a thin carbon film. Grids were plunge-frozen in liquid ethane using a Mark IV Vitrobot (ThermoFisher) at 100% humidity after blotting for 3–6 s. The images were recorded with a Gatan K2 direct electron detector in a Krios electron microscope in counting mode at 300 kV. During the exposure the defocus was stepped from a low value (2–4 µm 40 e$^-$/Å exposure) to a high value (~10 µm 20–40 e$^-$/Å). The focal pairs of micrographs were merged for improved visibility of the protein for particle picking as described[60].

The cryo-EM structure of bovine mitochondrial ATP synthase was used as an initial reference for the particle picking and refinement in RELION-1.3[61]. Particles were picked after vesicle-membrane subtraction by using the procedures described in[60]. Making use of the RELION rlnReconstructImageName variables, in the last iteration of 3D auto-refinement in RELION-1.3 an un-subtracted particle stack was used for the final reconstruction. This resulted in 3D reconstructions that included membrane density.

A 96-pixel box size (2.8 Å/pixel) was used for initial particle picking. Classification in 2D of picked particles was performed in RELION-1.3, and 7795 particles were used for reconstruction of the first 3D density map, which had 19 Å resolution. Subsequently, a larger box (128-pixel box, 4.2 Å/pixel) was used for particle extraction, chosen to fit ATP synthase dimers. Reconstruction of the second 3D map from 6655 particles resulted in 30 Å resolution. For this refinement the first reconstruction was used as a reference (filtered to 60 Å). Models were fitted with the atomic model of bovine ATP synthase (5ARA) by using UCSF Chimera[62].

**LC-MS/MS analysis.** In-solution protein digestion: Purified ATP synthase solution (5 µg/µl) in 0.1% DDM or liposome reconstituted ATP synthase solution (5 µg/µl) was precipitated with acetone by using established protocols to remove detergent. Protein pellet was dissolved and denatured in 8 M urea, 0.4 M ammonium bicarbonate, pH 8. The protein was reduced by the addition of 1/10 volume of 45 mM dithiothreitol (Pierce Thermo Scientific, #20290) and incubation at 37 °C for 20 min. Then the sample was alkylated with the addition of 1/10 volume of 100 mM iodoacetamide (Sigma-Aldrich, #I1149) with incubation in the dark at room temperature for 20 min. The urea concentration was adjusted to 2 M by the addition of water prior to enzymatic digestion at 37 °C with chymotrypsin for 16 h. Protease:protein ratios were estimated at 1:50. The sample was acidified by the addition of 20% trifluoroacetic acid, then desalted by using C18 MacroSpin columns (The Nest Group, #SMM SS18V) following the manufacturer's directions with peptides eluted with 0.1% TFA, 80% acetonitrile. Eluted sample was speedvac dried and dissolved in MS loading buffer (2% acetonitrile, 0.2% trifluoroacetic acid). A nanodrop measurement (Thermo Scientific Nanodrop 2000) determined protein concentrations (A260/A280). Each sample was then further diluted with MS loading buffer to 0.08 µg/µl, with 0.4ug (5 µl) injected for LC-MS/MS analysis.

LC-MS/MS on the Thermo Scientific Orbitrap Fusion: LC-MS/MS analysis was performed on a Thermo Scientific Orbitrap Fusion equipped with a Waters nanoAcquity UPLC system utilizing a binary solvent system (Buffer A: 100% water, 0.1% formic acid; Buffer B: 100% acetonitrile, 0.1% formic acid). Trapping was performed at 5 µl/min, 99.5% Buffer A for 3 min using a Waters Symmetry® C18 180 µm × 20 mm trap column. Peptides were separated using an ACQUITY UPLC PST (BEH) C18 nanoACQUITY Column 1.7 µm, 75 µm × 250 mm (37 °C) and eluted at 300 nl/min with the following gradient: 3% buffer B at initial conditions; 6% B at 5 minute; 35% B at 90 min; 85% B at 95 min; 85% B at 105 min; return to initial conditions at 106–120 min. MS was acquired in the Orbitrap in profile mode over the 300–1500 m/z range using quadrupole isolation, 1 microscan, 120,000 resolution, AGC target of $4 \times 10^5$, and a maximum injection time of 60 ms. MS/MS were collected in top speed mode with a 3 s cycle time on species with an intensity threshold of $5 \times 10^4$, charge states 2–8, peptide monoisotopic precursor selection preferred. Dynamic exclusion was set to 30 s. Data dependent MS/MS were acquired in the Orbitrap in centroid mode using quadrupole isolation (window 1.6 m/z), HCD activation with a collision energy of 28%, 1 microscan, 60,000 resolution, AGC target of $1 \times 10^5$, maximum injection time of 110 ms.

For the peptide identification, data were analyzed by using Byonic Software (Protein Metrics, San Carlos, CA USA; version PMI-Byonic-Demo:v2.15.7). Byonic was set up to search the data against the UniProt database with taxonomy restricted to Sus Scrofa (the domestic pig), (40,706 proteins). Search parameters included chymotrypsin digestion with up to two missed cleavages, peptide mass tolerance of 10 ppm, MS/MS fragment tolerance of 25 ppm, and methionine oxidation and carbamidomethylation on cysteine as variable modifications. The estimated-spectrum level FDR (false discovery rate) on true proteins was set to 0.3%. Scaffold (version Scaffold_4.8.9, Proteome Software) was used to validate and compare MS/MS based peptide and protein identifications. Normal and decoy database searches were run, with the confidence level set to 95% ($p < 0.05$).

Protein identifications were accepted if they could be established at greater than 99.0% probability and contained at least two identified peptides. Scaffold was also used for quantitative analysis of proteins to calculate the exponentially modified protein abundance index (emPAI).

**Statistical analysis.** Paired or unpaired Student's two-tailed $t$ test was used for two group comparisons. One-way ANOVA was used for multiple analysis, $p$ values are provided in figure legends. Data represented as mean ± SEM.

**Reporting summary.** Further information on research design is available in the Nature Research Reporting Summary linked to this article.

## Data availability
Data supporting the findings of this manuscript are available from the corresponding authors upon reasonable request. A reporting summary for this article is available as a Supplementary Information file. The source data underlying Fig. 1b, c, d, 3c, h, 4c, d, 5d and Supplementary Figs 1a, b, d, 3d are provided as a Source Data file. ATP synthase models from the Fig. 2a, e have been deposited into the Electron Microscopy Data Bank (accession codes EMD-21001, EMD-21002, respectively). The mass spectrometry proteomics data have been deposited to the ProteomeXchange Consortium via the PRIDE partner repository[63] with the dataset identifier PXD016255.

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

## Acknowledgements

We thank Dr Leonard K. Kaczmarek for insightful scientific discussion, Dr Shenping Wu for invaluable assistance with electron microscopy imaging, Besnik Murtishi for valuable technical assistance. We also would like to thank Jean Kanyo and Dr TuKiet Lam for their assistance with LC-MS/MS analysis of samples. Work supported by NIH Grant NS064967 (to E.A.J.), NIA Grant K01AG054734 (to N.M.).

## Author contributions

Performed the experiments: N.M., M.L., analyzed the data: N.M., E.A.J., F.J.S. Contributed reagents/materials/analysis tools, protocols: Y.Y., Y.Y., J.W. Writing-original draft: N.M., writing-review & editing: N.M., E.A.J., F.J.S., J.W., M.L.

## Competing interests

The authors declare no competing interests.
