## [Peer Review File · Nature Communications]

Reviewers' comments:

Reviewer #1 (Remarks to the Author):

Mnatsakanyan and coworkers report that the membrane bound proton channel of monomeric mitochondrial F-ATP synthase acts as a voltage gated ion channel with properties reminiscent of the elusive mitochondrial permeability transition pore (mPTP). The nature of the mPTP is highly controversial, with several studies implicating the dimeric F₁F₀ ATP synthase and/or the c subunits of the F₀ to act as the mPTP, and others providing evidence that excludes any such role for the enzyme in the process. Here the authors purify and reconstitute the monomeric enzyme into small vesicles and they use cryo-EM to verify that the enzyme is indeed monomeric. That part of the work is very well done, and while they do not provide a figure to support their claim of a 16 Å resolution, this number is of little significance to the authors' main conclusion. The authors then fuse the monomeric enzyme containing small vesicles with giant unilamellar vesicles (GUVs) for patch clamp single channel electrophysiology recordings. From the data, the authors conclude that monomeric ATP synthase possesses a voltage dependent conductance of ~1.3 nS, with the conductance and gating behavior being dependent on Ca²⁺ and ATP, characteristic modulators of the mPTP. They also show convincingly (using western analysis) that ANT and CypD are absent from the preparation.

The nature of the mPTP is an important question, and providing convincing and conclusive evidence that ATP synthase is - or is not - the mPTP would be a major contribution. However, it is unclear whether the current study can put the issue to rest.

The authors have already shown (ref. 6) that the purified c subunits of the F₀ display a conductance of ~1-2 nS that is sensitive to ATP and Ca²⁺, and since monomeric ATP synthase has the same c subunit ring, it does not seem all that surprising that they find similar ion conducting properties.

It is also not considered that F₁ could be lost during reconstitution, so that both F₀ and F₁F₀ complexes are present in the GUVs. This is a question they could address experimentally and that may help with solving the mPTP riddle - stripping F₁ and see whether the conductance persists.

Also, they cite Fig. 1e as evidence that incorporation of F₁F₀ is unidirectional. While some F₁ like densities are indeed seen at the periphery of the vesicles, some are seen to overlap the interior of the vesicles (for example the prominent vesicle in the middle of the lower half of the figure), and it cannot be known for sure whether these densities are pointing into the lumen, or towards the outside of the vesicle. Directionality of reconstitution could be tested by ATPase activity +/- detergent, or by sensitivity to protease digestion, or modification by water soluble fluorescent labels, etc.

Moreover, the authors make little attempt at providing a possible molecular mechanism that can rationalize the various observations. It must be assumed that the central pore of the c subunit ring (presumably c₈) is filled with lipid molecules to prevent a short circuit (conductance of protons) during e.g. state 3 respiration. Are the authors speculating that upon activation of mPTP function of ATP synthase (or c subunit ring), lipids get ejected from the central pore of the c subunit ring to allow passage of ions? What would provide the necessary free energy to eject these lipids?

It is also unclear what the role of ATP could be. No information is given whether Mg²⁺ was present during the electrophysiology experiments - so it is unknown whether the channel inhibition could be due to MgATP binding and hydrolysis on F₁, or whether ATP blocks the c subunit central pore directly (as implied by the earlier experiments with purified c subunits only, ref. 6). A schematic drawing showing the electrophysiology setup would help the non-expert interpret the experiments and the potential mechanism of action of the various modulators of the channel conductance.

The authors cite mass spectrometry analysis as evidence for absence of subunits e and g. Mass spectrometry is a powerful tool to prove presence of proteins (peptides), and while these subunits might indeed be absent from the preparation of the monomeric enzyme, mass spectrometry can not prove absence of a particular protein (especially membrane proteins) unless control experiments with dimeric ATP synthase are performed under identical conditions.

And finally, the silver stained gel of the preparation of monomeric ATP synthase (Fig. 1c) shows numerous contaminating bands (e.g. above alpha and below beta, below 37 kDa etc.). Without knowledge of the nature of these bands, it is hard to say whether there could be contaminating ion conducting channel or transporter proteins that could at least in part be responsible for the observed conductance profiles. These bands could be cut from the gel and analyzed separately.

Reviewer #2 (Remarks to the Author):

Mnatsakanyan and colleagues make a significant contribute to the literature pertaining to the role of ATP synthase in the mitochondrial permeability transition (mPTP). The state of that literature is a mess. While the current manuscript does not remove the confusion, it does provide important balance for the different sides in the ongoing controversy.

From the perspective of an interested outsider, the current state of the field looks like this: the Bernardi group in Padua has argued that the mitochondrial permeability transition pore is the dimeric ATP synthase. The Walker group in Cambridge has deleted ATP synthase subunits and argues in the literature that if the PT is due to ATP synthase, it could only be from the subunits e and g (which are found in yeast dimers and bovine monomers; according to the current paper not in porcine monomers). They may this statement because the phenomenon persists in cultured cells even after knocking out numerous subunits of ATP synthase (subunits a, 8 c, OSCP have reportedly been eliminated). They also say (but haven't published – as far as I am aware) that they've deleted subunits e and g and the PT still occurs in cultured cells. Further, from talking to experts in cellular bioenergetics, it apparently remains controversial whether or not the permeability transition observed in vitro is physiologically relevant.

The experiment performed in this concise manuscript is the reconstitution (with supporting cryoEM) of porcine ATP synthase and electrophysiological measurements with the resulting proteoliposomes. Both lines of investigation appear to be well done, with the cryoEM providing a low-resolution structure of monomeric ATP synthase in a liposome. By itself, the cryoEM structure does not add much structural knowledge: higher-resolution structures are available for quite a few different ATP synthases from single particle EM in detergents and lipid nanodiscs, as well as comparable resolution from electron tomography of sub-mitochondrial particles. The main significance of the EM work is to show that the ATP synthase was indeed reconstituted as a monomeric complex, which is not surprising considering the detergent used. However, the use of single particle methods with the ATP synthase in a proteoliposome is a nice technical accomplishment.

I am surprised that subunits e and g were not identified in the preparation. Subunits e and g are typically removed from yeast ATP synthases by the detergent DDM but remain with the bovine mitochondrial enzyme – although they can be difficult to detect by mass spec or SDS-PAGE because they are hydrophobic. The SDS-PAGE gel shown does not prove that subunits e and g are indeed missing as there a multiple small unlabelled bands at the bottom of the gel that could be due to these two small proteins. Proving the absence of a protein is, of course, very hard to do. One bit of evidence could come from single particle EM (negative stain or cryoEM). ATP synthase particles with subunits e and g (ie. seen in the bovine enzyme) have a characteristic 'bent' Fo region while particles without e and g (ie. seen in the yeast enzyme) have a round Fo region.

The authors describe (p. 4, line 122) an F1-c10 structure into their map. This cannot be a

complete description of what they have done as Fig. 2c shows images that include the stator, which is not present in F1-c10. Presumably they fit a yeast structure because the bovine structure, which includes subunits e and g and has a bent membrane region as a result, did not fit. However, it is not clear exactly what was fit or why.

I am also confident in the quality of the electrophysiology experiments, considering the expertise of the authors.

Thus, the major contribution from this manuscript is the statement that a permeability transition in vitro (whatever its physiological significance) can be induced by monomeric ATP synthase (that, in my opinion, may or may not contain subunits e and g). If the authors' enzyme preparation really does lack subunits e and g, their statement could be in conflict with the findings from the Walker group, who through knockout experiments concluded that only e and g could be the cause of the phenomenon if it is indeed due ATP synthase (a hypothesis they clearly do not think is true). It is also in conflict with the findings of the Bernardi group, who concluded that the channel occurs at the interface between monomers in the ATP synthase dimer. The Bernardi group purified dimers (through use of digitonin) and reconstituted them – but here the authors argue that they did not show their protein remained as dimers after reconstitution, which seems unlikely as in the hands of the Kuhlbrandt group reconstituted dimers in digitonin remain as dimers.

Thus, although the current manuscript does not provide a resolution to the controversies about the PTP that continue to confuse interested observers of the field, it does provide an important observation: that reconstitution of monomeric ATP synthase can lead to similar electrophysiology results seen with reconstitution of dimers.

I hesitate to get involved in this debate because I have not formed a strong opinion about what is going on. Nonetheless, in the spirit of scientific openness I am signing this report.

Reviewer #3 (Remarks to the Author):

The paper by Mnatsakanyan in a straightforward way supports previous claims of the research group that MONOMERIC ATP synthase is sufficient for mPTP activity. This view is in opposition to several high-quality papers of Paulo Bernardi and collaborators in which it was shown that the DIMERS not monomers of ATP synthase are specifically responsible for mPTP activity. With this respect, several control experiments should be included in the paper to further substantiate the author's evidence.

Major points include:

- 1) The purity of the enzyme preparation should be addressed in more detail. Fig1.C shows clearly that assigned subunits of ATP synthase make up most of the protein content. However, the preparation contains proteins (or their fragments?), which bands have been described/assigned neither in this figure nor in Table S1. Are all proteins detected by mass spectrometry included in this table? In addition, it is noticeable that a low number of peptides from proteins of Fo subunit was detected by mass spectrometry (probably due to their hydrophobic nature). Therefore, to further support the claim that no e and g subunits are present in the preparation, Western blots with specific antibodies against these two subunits should also be carried out.
- 2) ATP synthase was solubilized with DDM prior to purification and reconstitution. For reconstitution prolonged dialysis was used. However, due to its high micellar size and low CMC, DDM is considered to be very difficult to remove by this method. Residual detergent might potentially be a reason for abnormal protein behavior in patch-clamp experiments. Why another method (e.g. adsorption on Biobeads SM-2) was not used for this task? If possible, reconstitution

by extensive detergent adsorption followed by patch-clamp recordings should be performed to confirm the behavior of monomers of ATP synthase in a complete absence of detergent.

3) Only Ca^{2+} and ATP were used in this study as pharmacological modulators of mPTP. However, pharmacology of the megachannel is much broader and should be here also addressed.

Specifically, it was shown that Ba^{2+} and Mg^{2+} act antagonistically to Ca^{2+} and inhibit megachannel activity. Gd^{3+} is also a known blocker of PTP. In addition, protons are known inhibitors of mPTP and it was suggested that unique histidine in OSCP subunit mediates this inhibition. It was also claimed that glyoxal modification of conserved Arg107 of subunit g leads to modification of mPTP activity. With the respect of authors claim that no dimerization hence the presence of g subunit is necessary for mPTP activity, it would be worth to check sensitivity to glyoxal of the purified monomeric ATP synthase and compare it to the sensitivity of mPTP activity from native membranes (mitoplast) derived from porcine heart mitochondria.

Minor points:

1) In Fig 4a, b - it is unclear why closed and open levels (c and o, respectively, red lines) were put exactly where they are. The open level does not superimpose any channel opening in the traces, while closed level does not superimpose closed/baseline level, even in the presence of ATP where channel remains virtually closed. If they were places just as reference marks to indicate the direction of channel opening it should be clearly stated.

2) In Fig 4d - NPo for the channels in the "control" conditions (without Ca^{2+}) seems to be unusually high (> 0.5). Recordings of the native mPTP under such conditions showed very low - close to zero open probability. In agreement with this seems to be the recording in Fig. 4a (before the addition of 0.5 mM Ca^{2+}) where open probability looks to be much lower than 0.5. Is this relatively high NPo because of the way it was calculated? This does not seem to be explained properly in the text.

The authors take advantage of previously (2016) published protocol for purification of monomers of ATP synthase lacking e and g proteins (reference 35 in the manuscript) and with high confidence showed that such preparation exhibits mPTP-like activity. Although the paper does not bring a new idea to the field I think that the answer for the specific question, whether MONOMER or DIMER of ATP synthase is responsible for mPTP activity, is crucial for the elucidation of the location and ultimately the structure of the mPTP pore - the mystery which remains unsolved for three decades. Therefore, I strongly support publication of this work after major revision.

We heartily thank the reviewers for their excellent comments that we believe have led to great improvements in the manuscript.

We address each comment individually below and we have highlighted all the changes in the manuscript text file.

Reviewer #1 (Remarks to the Author):

Mnatsakanyan and coworkers report that the membrane bound proton channel of monomeric mitochondrial F-ATP synthase acts as a voltage gated ion channel with properties reminiscent of the elusive mitochondrial permeability transition pore (mPTP). The nature of the mPTP is highly controversial, with several studies implicating the dimeric F₁F₀ ATP synthase and/or the c subunits of the F₀ to act as the mPTP, and others providing evidence that excludes any such role for the enzyme in the process. Here the authors purify and reconstitute the monomeric enzyme into small vesicles and they use cryo-EM to verify that the enzyme is indeed monomeric. **That part of the work is very well done, and while they do not provide a figure to support their claim of a 16 Å resolution, this number is of little significance to the authors' main conclusion.**

We included the Fourier shell correlation (FSC) curve, which shows the resolution of the cryo-EM map (see Supplementary Figure 1c). We performed 3D reconstruction again for the ATP synthase reconstituted liposome to also include the membrane region, as suggested by the second reviewer. The resolution for the new image is 19 Å.

The authors then fuse the monomeric enzyme containing small vesicles with giant unilamellar vesicles (GUVs) for patch clamp single channel electrophysiology recordings. From the data, the authors conclude that monomeric ATP synthases possesses a voltage dependent conductance of ~1.3 nS, with the conductance and gating behavior being dependent on Ca²⁺ and ATP, characteristic modulators of the mPTP. They also show convincingly (using western analysis) that ANT and CypD are absent from the preparation. The nature of the mPTP is an important question, and providing convincing and conclusive evidence that ATP synthase is - or is not - the mPTP would be a major contribution. However, it is unclear whether the current study can put the issue to rest.

The authors have already shown (ref. 6) that the purified c subunits of the F₀ display a conductance of ~1-2 nS that is sensitive to ATP and Ca²⁺, and since monomeric ATP synthase has the same c subunit ring, it does not seem all that surprising that they find similar ion conducting properties.

It is also not considered that F₁ could be lost during reconstitution, so that both F₀ and F₁F₀ complexes are present in the GUVs. This is a question they could address experimentally and that may help with solving the mPTP riddle - stripping F₁ and see whether the conductance persists.

We have now included the immunoblot analysis of ATP synthase after reconstitution (see Supplementary Figure 1b), which confirms that the loss of F₁ does not occur in ATP synthase reconstituted GUVs. We have included Mg²⁺ and ATP (which bind to F₁) in all our buffers during protein purification and reconstitution to ensure stability and full assembly of ATP synthase.

To follow the reviewer's suggestion, we treated liposomes with sodium bromide to completely strip F₁. Immunoblot analysis showing complete depletion of F₁ in this sample is included in Supplementary figure 1b. We used this F₁-stripped preparation for patch-clamp recordings, and we demonstrate that the channel conductance persists under these conditions (Supplementary Figure 3b). We also show that Ba²⁺ does not inhibit channel activity in F₁-stripped samples (Supplementary Figure 3b). We have characterized the channel activity of purified reconstituted c-subunit, as well as of F₁-stripped submitochondrial vesicles (urea-treated SMVs) in our previous publication (Alavian et al., 2014). We have also shown that Ca²⁺-sensitivity is abolished in F₁-stripped SMVs, suggesting that the binding sites for Ca²⁺ and Ba²⁺ are located in F₁.

Also, they cite Fig. 1e as evidence that incorporation of F₁F₀ is unidirectional. While some F₁ like densities are indeed seen at the periphery of the vesicles, some are seen to overlap the interior of the vesicles (for example the prominent vesicle in the middle of the lower half of the figure), and it cannot be known for sure whether these densities are pointing into the lumen, or towards the outside of the vesicle. Directionality of reconstitution could be tested by ATPase activity

+/- detergent, or by sensitivity to protease digestion, or modification by water soluble fluorescent labels, etc.

Immunoblot analysis of sodium bromide treated liposomes shows complete absence of F₁ alpha subunit after treatment (Supplementary figure 1b). Based on this we suggest that reconstitution is unidirectional since sodium bromide can only remove F₁ if it is outside but not inside the liposomes. It is therefore likely that the images shown in Figure 1e are the top views of reconstituted ATP synthase.

Moreover, the authors make little attempt at providing a possible molecular mechanism that can rationalize the various observations. It must be assumed that the central pore of the c subunit ring (presumably c8) is filled with lipid molecules to prevent a short circuit (conductance of protons) during e.g. state 3 respiration. Are the authors speculating that upon activation of mPTP function of ATP synthase (or c subunit ring), lipids get ejected from the central pore of the c subunit ring to allow passage of ions? What would provide the necessary free energy to eject these lipids?

We have now included more intense discussion about the possible molecular mechanisms of channel opening (see Discussion).

It is also unclear what the role of ATP could be. No information is given whether Mg²⁺ was present during the electrophysiology experiments - so it is unknown whether the channel inhibition could be due to MgATP binding and hydrolysis on F₁, or whether ATP blocks the c subunit central pore directly (as implied by the earlier experiments with purified c subunits only, ref. 6).

We have used disodium and not magnesium salt of ATP during recordings; therefore, we have ATP-dependent inhibition of the channel activity. We have updated the manuscript with this information. We have previously published that there are two distinct ATP binding sites, a high affinity site located in F₁ and a low affinity site located in c-subunit (Alavian et al, 2014).

A schematic drawing showing the electrophysiology setup would help the non-expert interpret the experiments and the potential mechanism of action of the various modulators of the channel conductance.

We have added this new image in Figure 3 (see Figure 3a, b) to address this concern.

The authors cite mass spectrometry analysis as evidence for absence of subunits e and g. Mass spectrometry is a powerful tool to prove presence of proteins (peptides), and while these subunits might indeed be absent from the preparation of the monomeric enzyme, mass spectrometry cannot prove absence of a particular protein (especially membrane proteins) unless control experiments with dimeric ATP synthase are performed under identical conditions.

We very much appreciate this comment. Western blot analysis showed the partial presence of e (~30%) and g (~50%) subunits in our preparation (see Supplemental Figure 1d for western blot analysis). We have now included these subunits in the silver stained gel subunit list (Fig. 1c) and added a figure for immunoblot analysis (Fig. S1d).

And finally, the silver stained gel of the preparation of monomeric ATP synthase (Fig. 1c) shows numerous contaminating bands (e.g. above alpha and below beta, below 37 kDa etc.). Without knowledge of the nature of these bands, it is hard to say whether there could be contaminating ion conducting channel or transporter proteins that could at least in part be responsible for the observed conductance profiles. These bands could be cut from the gel and analyzed separately.

We thank the reviewer for this comment. We cut out the bands and analyzed them with LC-MS/MS, which showed that the contaminating bands are the respiratory chain Complex 1 subunits 1 (80 kDa) and subunit 2 (54 kDa). It has been reported that complex 1 does not form a voltage-gated channel (Giorgio et al, 2013). We showed a silver stain gel of purified ATP synthase in the previous version of manuscript, now we replace it with the silver stain gel of the ATP synthase sample after reconstitution into liposomes to demonstrate the proteins that are present during recordings.

We have also included the raw data of LC-MS/MS with the quantitative analysis of proteins (Supplementary Table 1). We used the scaffold software for quantitative analysis of proteins to calculate their exponentially modified protein abundance

index (emPAI). Proteins are listed based on their emPAI values. Even though ANT1 was not identified by immunoblot analysis (Supplementary figure 1a), the LC-MS/MS analysis showed a residual contamination with ANT1 (0.54% relative to ATP synthase β subunit abundance) but not with CypD, VDAC or PiC.

In this revised version of manuscript, we now also show that ATP synthase channel activity can be inhibited by the ATP synthase specific inhibitor oligomycin, which strongly suggests that ATP synthase is solely responsible for the channel activity recorded in our studies.

Reviewer #2 (Remarks to the Author):

Mnatsakanyan and colleagues make a significant contribute to the literature pertaining to the role of ATP synthase in the mitochondrial permeability transition (mPTP). The state of that literature is a mess. While the current manuscript does not remove the confusion, it does provide important balance for the different sides in the ongoing controversy.

From the perspective of an interested outsider, the current state of the field looks like this: the Bernardi group in Padua has argued that the mitochondrial permeability transition pore is the dimeric ATP synthase. The Walker group in Cambridge has deleted ATP synthase subunits and argues in the literature that if the PT is due to ATP synthase, it could only be from the subunits e and g (which are found in yeast dimers and bovine monomers; according to the current paper not in porcine monomers). They may this statement because the phenomenon persists in cultured cells even after knocking out numerous subunits of ATP synthase (subunits a, 8 c, OSCP have reportedly been eliminated). They also say (but haven't published – as far as I am aware) that they've deleted subunits e and g and the PT still occurs in cultured cells. Further, from talking to experts in cellular bioenergetics, it apparently remains controversial whether or not the permeability transition observed in vitro is physiologically relevant.

The experiment performed in this concise manuscript is the reconstitution (with supporting cryoEM) of porcine ATP synthase and electrophysiological measurements with the resulting proteoliposomes. Both lines of investigation appear to be well done, with the cryoEM providing a low-resolution structure of monomeric ATP synthase in a liposome. By itself, the cryoEM structure does not add much structural knowledge: higher-resolution structures are available for quite a few different ATP synthases from single particle EM in detergents and lipid nanodiscs, as well as comparable resolution from electron tomography of sub-mitochondrial particles. The main significance of the EM work is to show that the ATP synthase was indeed reconstituted as a monomeric complex, which is not surprising considering the detergent used. However, the use of single particle methods with the ATP synthase in a proteoliposome is a nice technical accomplishment.

I am surprised that subunits e and g were not identified in the preparation. Subunits e and g are typically removed from yeast ATP synthases by the detergent DDM but remain with the bovine mitochondrial enzyme – although they can be difficult to detect by mass spec or SDS-PAGE because they are hydrophobic. The SDS-PAGE gel shown does not prove that subunits e and g are indeed missing as there are multiple small unlabeled bands at the bottom of the gel that could be due to these two small proteins. Proving the absence of a protein is, of course, very hard to do. One bit of evidence could come from single particle EM (negative stain or cryoEM). ATP synthase particles with subunits e and g (ie. seen in the bovine enzyme) have a characteristic ‘bent’ Fo region while particles without e and g (ie. seen in the yeast enzyme) have a round Fo region.

As we stated above the quantification of western blot analysis showed the partial presence of e (~30%) and g (~50%) subunits in our preparation (see Supplemental Figure 1d for western blot analysis).

Even though western blot analysis showed that the e and g subunits are partially present in the sample, the ATP synthase was present in its monomeric conformation based on our cryo-EM data. We performed 3D reconstruction again for the ATP synthase reconstituted liposomes to also include the membrane

region, the 'bent' region of the membrane was not obvious in either one of our maps.

The authors describe (p. 4, line 122) an F1-c10 structure into their map. This cannot be a complete description of what they have done as Fig. 2c shows images that include the stator, which is not present in F1-c10. Presumably they fit a yeast structure because the bovine structure, which includes subunits e and g and has a bent membrane region as a result, did not fit. However, it is not clear exactly what was fit or why.

We sincerely apologize for this mistake. We have used the cryo-EM map of bovine monomeric ATP synthase for fitting (5ARA, Zhou et al, 2015). We have corrected this in the manuscript.

I am also confident in the quality of the electrophysiology experiments, considering the expertise of the authors.

Thus, the major contribution from this manuscript is the statement that a permeability transition in vitro (whatever its physiological significance) can be induced by monomeric ATP synthase (that, in my opinion, may or may not contain subunits e and g). If the authors' enzyme preparation really does lack subunits e and g, their statement could be in conflict with the findings from the Walker group, who through knockout experiments concluded that only e and g could be the cause of the phenomenon if it is indeed due ATP synthase (a hypothesis they clearly do not think is true). It is also in conflict with the findings of the Bernardi group, who concluded that the channel occurs at the interface between monomers in the ATP synthase dimer. The Bernardi group purified dimers (through use of digitonin) and reconstituted them – but here the authors argue that they did not show their protein remained as dimers after reconstitution, which seems unlikely as in the hands of the Kuhlbrandt group reconstituted dimers in digitonin remain as dimers.

Here we suggest that the dimer formation is not required for the channel conductance and that the channel is located within the monomer. This does not mean that the ATP synthase dimers cannot form a channel, two channels instead

of one should be located within a dimer. In addition, our findings do not preclude the existence of other mitochondrial inner membrane channels that can depolarize the inner membrane upon Ca^{2+} overload. Our study only shows that the ATP synthase houses a very large and abundant channel that is most likely the main contributor to the “mPTP”.

Thus, although the current manuscript does not provide a resolution to the controversies about the PTP that continue to confuse interested observers of the field, it does provide an important observation: that reconstitution of monomeric ATP synthase can lead to similar electrophysiology results seen with reconstitution of dimers.

I hesitate to get involved in this debate because I have not formed a strong opinion about what is going on. Nonetheless, in the spirit of scientific openness I am signing this report.

Reviewer #3 (Remarks to the Author):

The paper by Mnatsakanyan in a straightforward way supports previous claims of the research group that MONOMERIC ATP synthase is sufficient for mPTP activity. This view is in opposition to several high-quality papers of Paulo Bernardi and collaborators in which it was shown that the DIMERS not monomers of ATP synthase are specifically responsible for mPTP activity. With this respect, several control experiments should be included in the paper to further substantiate the author's evidence. Major points include:

1) The purity of the enzyme preparation should be addressed in more detail. Fig1.C shows clearly that assigned subunits of ATP synthase make up most of the protein content. However, the preparation contains proteins (or their fragments?), which bands have been described/assigned neither in this figure nor in Table S1. Are all proteins detected by mass spectrometry included in this table? In addition, it is noticeable that a low number of peptides from proteins of Fo subunit was detected by mass spectrometry (probably due to their hydrophobic nature). Therefore, to further support the claim that no e and g

subunits are present in the preparation, Western blots with specific antibodies against these two subunits should also be carried out.

As we mentioned above, we have included the LC-MS/MS data with the quantitative analysis of proteins (Supplementary Table 1), as well as western blots for e and g subunits (Supplemental Figure 1d).

2) ATP synthase was solubilized with DDM prior to purification and reconstitution. For reconstitution prolonged dialysis was used. However, due to its high micellar size and low CMC, DDM is considered to be very difficult to remove by this method. Residual detergent might potentially be a reason for abnormal protein behavior in patch-clamp experiments. Why another method (e.g. adsorption on Biobeads SM-2) was not used for this task? If possible, reconstitution by extensive detergent adsorption followed by patch-clamp recordings should be performed to confirm the behavior of monomers of ATP synthase in a complete absence of detergent.

We have now used SM-2 Bio-Beads during GUV preparation to further remove the residual detergent if it was present. We have used these newly prepared GUVs for patch-clamp recordings to characterize the channel (see Figure 5, Supplementary Figure 3 for recordings).

3) Only Ca^{2+} and ATP were used in this study as pharmacological modulators of mPTP. However, pharmacology of the megachannel is much broader and should be here also addressed. Specifically, it was shown that Ba^{2+} and Mg^{2+} act antagonistically to Ca^{2+} and inhibit megachannel activity. Gd^{3+} is also a known blocker of PTP. In addition, protons are known inhibitors of mPTP and it was suggested that unique histidine in OSCP subunit mediates this inhibition. It was also claimed that glyoxal modification of conserved Arg107 of subunit g leads to modification of mPTP activity. With the respect of authors claim that no dimerization hence the presence of g subunit is necessary for mPTP activity, it would be worth to check sensitivity to glyoxal of the purified monomeric ATP synthase and compare it to the sensitivity of mPTP activity from native membranes (mitoplast) derived from porcine heart mitochondria.

We have now used several known pharmacological modulators of mPTP to check their effect on ATP synthase channel activity. We have added these recordings to Figure 5, which shows that the channel can be inhibited with Ba^{2+} and Gd^{3+} . We now also show that Ba^{2+} inhibition occurs in a reversible and F_1 -dependent manner (Supplementary Figure 3). The addition of Ba^{2+} (1mM) inhibits the channel, which reopens after washing the patch with recording buffer. During the same recording the channel closes again after subsequent addition of Ba^{2+} (1 mM). Additionally, we find that Ba^{2+} fails to close the channel during continuous recordings of liposomes with fully stripped F_1 (Supplementary Figure 3), indicating that the binding site for Ba^{2+} is located in F_1 .

Most importantly, we now definitively demonstrate that the channel is closed by the ATP synthase specific inhibitor oligomycin (Figure 5), therefore we suggest, given the findings that Ca^{2+} , Ba^{2+} and ATP bind within F_1 , that the oligomycin induced conformational changes in c-subunit transfer to F_1 , producing channel inactivation through the tighter coupling of F_1 with F_0 .

Unlike the effect of Ba^{2+} , Gd^{3+} and oligomycin, where we had immediate inhibition with 100% success in all patches, with phenylglyoxal (PGO) we could only see a brief, partial inhibition, which was observed only in two patches out of six. We therefore chose not to show these data.

Binding of different synthetic PGO derivatives was shown to influence the opening and closing of the mPTP through the modulation of arginine residues, which were suggested to play a role as voltage sensors of mPTP (Johans M, 2005, JBC). The effect of PGO was also shown to be species-specific: it inhibits the mPTP in mouse and yeast whereas in contrast it induces it in human and drosophila mitochondria (Guo et al., 2018). As the reviewer mentions, the arginine residue R107 of yeast ATP synthase subunit g was recently found to be an mPTP modulating target of PGO in yeast (Guo et al., 2018). They find that the interaction between subunits e and g (and their specific arginine residues) in yeast F-ATP synthase is important for both dimer formation and for the full-conductance MMC/PTP. They explain the effect of glyoxal as working through the modification of arginine residues in subunits e and g which also interferes with dimer stability.

Glyoxal is a widely used and non-selective cross-linker that could have many binding sites within the porcine ATP synthase, given the presence of numerous arginine residues in ATP synthase. Therefore, based on this non-specific behavior of glyoxal, the fact that we have only ATP synthase monomers after reconstitution in liposomes and partial presence of subunits g and e in our sample, we find that any interpretation of glyoxal effect in the recordings in this present study would be too preliminary for publication.

Minor points:

1) In Fig 4a, b - it is unclear why closed and open levels (c and o, respectively, red lines) were put exactly where they are. The open level does not superimpose any channel opening in the traces, while closed level does not superimpose closed/baseline level, even in the presence of ATP where channel remains virtually closed. If they were placed just as reference marks to indicate the direction of channel opening it should be clearly stated.

We have updated all the figures to represent the fully closed state as C and fully open state as O.

2) In Fig 4d - NPo for the channels in the "control" conditions (without Ca²⁺) seems to be unusually high (> 0.5). Recordings of the native mPTP under such conditions showed very low - close to zero open probability. In agreement with this seems to be the recording in Fig. 4a (before the addition of 0.5 mM Ca²⁺) where open probability looks to be much lower than 0.5. Is this relatively high NPo because of the way it was calculated? This does not seem to be explained properly in the text.

We thank the reviewer for this comment. We have measured the NPo for all subconductance states and not only for the peak conductance levels, which made the value much higher. Now we explain this in the text.

The authors take advantage of previously (2016) published protocol for purification of monomers of ATP synthase lacking e and g proteins (reference 35 in the manuscript) and with high confidence showed that such preparation exhibits mPTP-like activity.

As it is mentioned above, we addressed this issue by performing western blot analysis with specific antibodies for subunits e and g (Supplemental Figure 1d).

Although the paper does not bring a new idea to the field I think that the answer for the specific question, whether MONOMER or DIMER of ATP synthase is responsible for mPTP activity, is crucial for the elucidation of the location and ultimately the structure of the mPTP pore - the mystery which remains unsolved for three decades. Therefore, I strongly support publication of this work after major revision.

We thank the reviewers for all the comments, which helped us to significantly improve the manuscript.

Reviewers' comments:

Reviewer #1 (Remarks to the Author):

In their revision, Mnatsakanyan et al. address many of the concerns from the first round of review satisfactorily, resulting in a much improved manuscript. However, some of the new data, and a few issues from the original manuscript, need further clarification.

(1) The authors convincingly show that monomeric holo ATP synthase and Fo subcomplex have voltage dependent ion channel activity that can be modulated by Ca²⁺ and blocked by ATP, oligomycin, Ba²⁺ and Gd³⁺. The question that remains, however, is whether this observed conductance is physiological, and if so, whether it's related to the elusive mPTP? In the authors' hands, monomeric ATP synthase (or the Fo) is constitutively open, as opposed to the physiological mPTP, that requires opening by e.g. Ca²⁺. How do the authors explain this discrepancy?

(2) The traces in Fig. 3 (especially panel d) are very different from the traces in subsequent figures 4 and 5 before adding Ca²⁺ or Ba²⁺/Gd³⁺. Traces also vary a lot within figures, e.g. Fig. 5a vs b. What are the possible reasons for the differences, and how might the differences affect interpretation of the measurements?

(3) The authors have added some discussion about the nature of the pore, but they are still not clearly saying what they think constitutes the pore. Do they think it's the central cavity of the c-ring? Or the polar interface between c-ring and the 'a' subunit? If it's the c-ring, would the diameter of the central pore be consistent with the observed conductance of up to 1.8 nS? The expected conductance of the central pore could be estimated from the cross section of the c-ring measured from crystal or cryoEM structures.

(4) At the time of the first round of review, the assumption was that the central pore of the c-ring is filled with lipid. However, a recent cryoEM structure (cited in the manuscript, ref. 54) shows that the central pore is occupied by a helical peptide (6.8PL). Do they see evidence of this small subunit in their mass spec analysis? Loss of this subunit during purification could explain why the channel is constitutively open in the authors' experiments.

(5) The authors now report that oligomycin inhibits the channel in case of the holo enzyme. This is curious as it is generally assumed that inhibition of F₁F_o is due to binding of oligomycin to the periphery of the c-ring, which in turn prevents rotation of the ring past subunit 'a'. However, the authors also now state that they did not include Mg²⁺ in their buffer (see rebuttal page 3), so even with ATP present, there would be no ATPase turnover. How then can inhibition of the channel be explained by oligomycin? Did the authors test whether oligomycin also inhibits the channel activity of liposomes stripped of F₁? Since oligomycin prevents c-ring rotation in free Fo, this experiment could tell them whether conductance is through the central c-ring pore, or via the subunit 'a' - c-ring interface.

(6) Page 18: This reviewer is not an expert in electrophysiology - but isn't conductance defined as the reciprocal of resistance? And since resistance $R = V/I$, shouldn't it be conductance $G = 1/V$ (and not $G = V/I$ as stated in the manuscript)? And consequently, are the conductance values reported in this manuscript calculated correctly? Please note that the same equation was also used in previous papers by the same group (e.g. ref. 6).

Reviewer #2 (Remarks to the Author):

Since the initial submission of this manuscript the Walker group in Cambridge has published a manuscript showing that any ATP synthase subunit can be knocked out without affecting the mPTP phenomenon in humans cells. Indeed, they see this phenomenon persist in the absence of any

assembled ATP synthase (Persistence of the permeability transition pore in human mitochondria devoid of an assembled ATP synthase, 2019, PNAS 116, 12816–12821).

The discrepancy between the cell-based and molecular studies is not addressed in the present manuscript. I have no reason to doubt the quality of the experiments described in the current manuscript or the manuscripts from the Walker group. Therefore, it is increasingly difficult to reconcile that the experiments described here with isolated molecules and that experiments showing a mPTP in human cells are actually reporting on the same phenomenon.

Reviewer #3 (Remarks to the Author):

Mnatsakanyan and coworkers positively addressed all the concerns of my review. These include reconstitution of the protein in the presence of SM-2 BioBeads, more extensive channel pharmacological profile and electrophysiological data presentation and analysis. Especially interesting seems to be the author's new observation that Ba^{2+} ions block the activity of the channel only in the presence of F1 subunit.

Authors also identified contaminant proteins in their preparation and included mass spectrometry data. However, in contrast to their previous claims, the preparation contains e and g subunits, although in reduced amounts (~30% and ~50%, respectively). This raises some important questions. Are e and g subunits associated with ATPase monomers in their preparation? Could e and g subunit presence lead to spontaneous transient (?) ATP synthase dimer formation? With this respect, cryo-EM experiments seem to be a crucial part of this work. Unfortunately, I am not competent in this field. In spite of this, I can think of one weak spot in the present manuscript. Authors analyze cryo-EM images of ATP synthase reconstituted into SUVs. However, to record channel activities the SUVs are fused with GUVs of different lipid composition (according to ref.6). There is only a very slight chance that such treatment could result in ATP synthase dimer formation, but unless it is shown, e.g. by cryo-EM, authors should be more careful with their statements. With this respect, I would suggest either to do additional cryo-EM analysis of the material that is directly used in patch-clamp experiments or to change the title and some statements for less definite.

In general, I find the paper after revision satisfactory and ready for publication with above-mentioned reservation.

We thank the reviewers for all the insightful comments which led to the improvement of the manuscript.

We address each comment individually below.

Reviewer 1.

(1) The authors convincingly show that monomeric holo ATP synthase and Fo subcomplex have voltage dependent ion channel activity that can be modulated by Ca²⁺ and blocked by ATP, oligomycin, Ba²⁺ and Gd³⁺. The question that remains, however, is whether this observed conductance is physiological, and if so, whether it's related to the elusive mPTP? In the authors' hands, monomeric ATP synthase (or the Fo) is constitutively open, as opposed to the physiological mPTP, that requires opening by e.g. Ca²⁺. How do the authors explain this discrepancy?

Response: We thank the reviewer for this comment. The mPTP has been repeatedly reported as a voltage-gated channel in previous studies (Petronilli et al, 1989, Szabo et al, 1992, Azzolin et al, 2010). In our current manuscript we show that the ATP synthase is not constitutively open, we observed the channel opening upon voltage application and the subsequent increase in channel activity in the presence of calcium. Additionally, we show that this channel is sensitive to known mPTP modulators: Ca²⁺, ATP, Ba²⁺ and Gd³⁺. Therefore, we show that the biophysical behavior of the ATP synthase channel is in good agreement with that of the mPTP.

(2) The traces in Fig. 3 (especially panel d) are very different from the traces in subsequent figures 4 and 5 before adding Ca²⁺ or Ba²⁺/Gd³⁺. Traces also vary a lot within figures, e.g. Fig. 5a vs b. What are the possible reasons for the differences, and how might the differences affect interpretation of the measurements?

Response: The channel is a multi-conductance channel, as has been reported for mPTP, with different open and closed levels but in addition to different conductance modes, it also has different frequency modes with different dwell times. This explains the different appearances of the channel activity in different figures.

(3) The authors have added some discussion about the nature of the pore, but they are still not clearly saying what they think constitutes the pore. Do they think it's the central cavity of the c-ring? Or the polar interface between c-ring and the 'a' subunit? If it's the c-ring, would the diameter of the central pore be consistent with the observed conductance of up to 1.8 nS? The expected conductance of the central pore could be estimated from the cross section of the c-ring measured from crystal or cryoEM structure.

Response: We thank the reviewer for this comment. In this paper we are not showing evidence that the c-subunit is the pore. We have published earlier (Alavian et al., 2014) a large study focused on the channel activity of the purified c-subunit. According to this study the pore diameter increases during channel opening, therefore we cannot accurately measure dynamic pore diameter based on currently available structures of the c-subunit.

(4) At the time of the first round of review, the assumption was that the central pore of the c-ring is filled with lipid. However, a recent cryoEM structure (cited in the manuscript, ref. 54) shows that the central pore is occupied by a helical peptide (6.8PL). Do they see evidence of this small subunit in their mass spec analysis? Loss of this subunit during purification could explain why the channel is constitutively open in the authors' experiments.

Response: We thank the reviewer for this comment. We are as keen to identify the gating mechanism of the ATP synthase channel as the reviewer; however, we cannot address this question in the scope of this manuscript.

As we hypothesized in the discussion, the 6.8 PL subunit could be displaced upon the application of voltage and play an important role in the channel gating mechanism. We only occasionally found 6.8 PL in our samples, since small hydrophobic proteins are challenging to find in mass spec analysis. As the authors report in ref 54, more studies are needed to definitively establish the identity of the protein occupying the center of the c-subunit ring.

This is a quote from reference 54 about the 6.8PL protein:

Page 2: "... we also identified a helical density in the center of the c₈-ring (fig. S12). Although we were unable to build the side chains of this subunit into the low-resolution density maps, **we propose** that this subunit may be the previously reported subunit 6.8PL for four reasons... Because our density map within the c₈-ring is not well resolved, we cannot exclude the possibility that this density belongs to a protein recruited into the c₈-ring when ATP synthase is inhibited. A structure with better defined density in this region will be needed to definitively establish the identity of this subunit."

The molecular mechanism of ATP synthase channel opening still needs to be discovered, we can only suggest different hypotheses based on the currently available ATP synthase structures.

(5) The authors now report that oligomycin inhibits the channel in case of the holo enzyme. This is curious as it is generally assumed that inhibition of F₁F_o is due to binding of oligomycin to the periphery of the c-ring, which in turn prevents rotation of the ring past subunit 'a'. However, the authors also now state that they did not include Mg²⁺ in their buffer (see rebuttal page 3), so even with ATP present, there would be no ATPase turnover. How then can inhibition of the channel be explained by oligomycin? Did the authors test whether oligomycin also inhibits the channel activity of liposomes stripped of F₁? Since oligomycin prevents c-ring rotation in free F_o, this experiment could tell them whether conductance is through the central c-ring pore, or via the subunit 'a' - c-ring interface.

Response: We thank the reviewer for such a perceptive comment. We do not have any evidence nor claim that the channel activity depends on ATP hydrolysis. We have already published that purified c-subunit has channel activity similar to the activity that we record in purified ATP synthase and again now show it in F₁-stripped ATP synthase. Whether or not F₁ is present, however, is unlikely to make a difference in oligomycin binding since the binding site for oligomycin is in F_o. The fact that we see oligomycin inhibition does not mean that we suggest that the channel is located at the interface of a and c subunits, where oligomycin binds. If that was the case, then we should not detect the channel activity with the purified c-subunit alone as we reported earlier (Alavian et al., 2014).

We think that oligomycin binding to F_0 induces conformational changes in ATP synthase structure to close the pore. We stated in the Discussion that future structural analysis of ATP synthase inhibited by oligomycin will perhaps reveal this conformation.

The lack of availability of a high-resolution structure for the F_0 subcomplex, consisting of various subunits, including the recently identified subunit occupying the center of the c-ring, will make it difficult to interpret the results of any oligomycin effect on F_1 -stripped ATP synthase on a molecular level. The goal for the oligomycin experiment was to show that the ATP synthase, rather than other possible contaminants present in the sample, was solely responsible for the observed channel activity.

(6) Page 18: This reviewer is not an expert in electrophysiology - but isn't conductance defined as the reciprocal of resistance? And since resistance $R = V/I$, shouldn't it be conductance $G = I/V$ (and not $G = V/I$ as stated in the manuscript)? And consequently, are the conductance values reported in this manuscript calculated correctly? Please note that the same equation was also used in previous papers by the same group (e.g. ref. 6).

Response: We thank the reviewer for this comment. The reviewer is correct and has noticed the typographical error, which we have corrected now to say $G=I/V$. The conductance values throughout the manuscript were calculated correctly, and the reviewer can also check the calculations simply by performing the calculations based on the information in the labeled traces.

Reviewer 2. Since the initial submission of this manuscript the Walker group in Cambridge has published a manuscript showing that any ATP synthase subunit can be knocked out without affecting the mPTP phenomenon in human cells. Indeed, they see this phenomenon persist in the absence of any assembled ATP synthase (Persistence of the permeability transition pore in human mitochondria devoid of an assembled ATP synthase, 2019, PNAS 116, 12816–12821).

The discrepancy between the cell-based and molecular studies is not addressed in the present manuscript. I have no reason to doubt the quality of the experiments described in the current manuscript or the manuscripts from the Walker group. Therefore, it is

increasingly difficult to reconcile that the experiments described here with isolated molecules and that experiments showing a mPTP in human cells are actually reporting on the same phenomenon.

Response: We make no claim in this paper that this channel is solely responsible for permeability transition, therefore both studies could be correct. As we mentioned in the Discussion, all of Dr. Walker's published experiments are indirect measurements of the mPTP. Dr. Walker's group did not perform ion channel recordings which are accepted as the gold standard for measuring ion channel activity. Our previous publication (Neginskaya et al, 2019) describes the measurements of ion channel activity of mitochondria from c-subunit knock out cells that we directly obtained from Dr. Walker's lab. Our results show that the higher conductance activity of mPTP is absent in these mitochondria and we therefore argue that the large conductance activity of mPTP is no longer present in these KO cells. Nevertheless, there is still present low conductance mitochondrial channel activity, which we attribute to the presence of many previously described inner membrane channels and transporters and which we assume can all contribute to membrane depolarization and mPTP induction in the absence of the largest "mPT" pore. Therefore, as we included in our discussion, many other low conductance channels/transporters exist in the inner mitochondrial membrane, but the largest conductance channel is located in the ATP synthase.

Reviewer 3.

Mnatsakanyan and coworkers positively addressed all the concerns of my review. These include reconstitution of the protein in the presence of SM-2 BioBeads, more extensive channel pharmacological profile and electrophysiological data presentation and analysis. Especially interesting seems to be the author's new observation that Ba²⁺ ions block the activity of the channel only in the presence of F1 subunit. Authors also identified contaminant proteins in their preparation and included mass spectrometry data. However, in contrast to their previous claims, the preparation contains e and g subunits, although in reduced amounts (~30% and ~50%,

respectively). This raises some important questions. Are e and g subunits associated with ATPase monomers in their preparation? Could e and g subunit presence lead to spontaneous transient (?) ATP synthase dimer formation? With this respect, cryo-EM experiments seem to be a crucial part of this work. Unfortunately, I am not competent in this field. In spite of this, I can think of one weak spot in the present manuscript. Authors analyze cryo-EM images of ATP synthase reconstituted into SUVs. However, to record channel activities the SUVs are fused with GUVs of different lipid composition (according to ref.6). There is only a very slight chance that such treatment could result in ATP synthase dimer formation, but unless it is shown, e.g. by cryo-EM, authors should be more careful with their statements. With this respect, I would suggest either to do additional cryo-EM analysis of the material that is directly used in patch-clamp experiments or to change the title and some statements for less definite. In general, I find the paper after revision satisfactory and ready for publication with above-mentioned reservation.

Response: We thank the reviewer for all the suggestions and comments.

We sincerely apologize for the confusion that we had in our first submission regarding e and g subunits. We could not initially detect e and g subunits in the mass spec analysis of our sample, due to the hydrophobicity of these subunits.

However later, thanks to reviewers' suggestions, these subunits were identified by the western blot analysis. We show the presence of e (~30%) and g (~50%) subunits in our preparation (see Supplemental Figure 1d).

It has been reported earlier by Dr. Rubinstein's group (Baker et al, 2012 PNAS), that the DDM-purified bovine ATP synthase still contains e and g subunits, but they are part of ATP synthase monomer according to the published cryo-EM structure in their above-mentioned manuscript (Please see below for figure 1 from their published paper). Here we also report the partial presence of e and g subunits in our samples, and at the same time we show the structure of ATP synthase monomer by cryo-EM. We have made our conclusion about the monomeric state of ATP synthase based on the cryo-EM analysis of purified reconstituted ATP synthase and not based on the presence or absence of e and g subunits. We could not detect the second ATP synthase molecule at the 86-degree angle position

required for dimer formation, even though e and g subunits were present, confirming that these subunits are part of the monomer, consistent with Baker et al., 2012 PNAS.

We understand the reviewer's concern about the different lipid composition used. As the reviewer has stated, there is only a very slight chance that the dimers will form during SUV to GUV fusion procedure. More importantly, in the larger vesicles with a less acute membrane curvature (GUVs), the possibility of forming dimers at the 86-degree angle required for dimer formation would be even less likely.

It is impossible to use GUVs for cryo-EM analysis, as the reviewer suggested, because of their gigantic size, but instead we performed additional patch-clamp recordings by preparing GUVs with the exact lipid composition that we used for SUV preparation. Now we show that ATP synthase reconstituted in GUVs with the lipid composition of SUVs that we used for cryo-EM imaging still forms a channel with the same peak conductance value and it is still sensitive to inhibition with barium (please see Fig. S3 c, d).

We heartily thank all the reviewers for their comments which have greatly improved the manuscript.

Supplementary figure 3.

Baker et al, 2012 PNAS, Figure 1. e and g subunits are found in the cryo-EM structure of ATP synthase monomer.

REVIEWERS' COMMENTS:

Reviewer #1 (Remarks to the Author):

In the second round of revision, the authors have addressed the critiques from the previous round of review satisfactorily and the manuscript can now be recommended for publication in Nature Communications.

Reviewer #3 (Remarks to the Author):

In general, I find the results and the author's responses to questions convincing and satisfactory. However, when this work has been under revision the paper by Urbani A, et al. (Nat Commun. 10(1):4341. doi: 10.1038/s41467-019-12331-1) was published in which it was shown that the channel activity associated with highly purified bovine ATP synthase is observed when dimers and tetramers but NOT monomers of this protein complex are fused with planar lipid membranes. This work is crucial and should be cited and discussed in the current manuscript. In particular, it should be discussed why Mnatsakanyan could observe channel activity of the monomeric ATP synthase while all the data from Bernardi group and his co-workers indicate that PTP activity resides only in ATP synthase dimers.

In spite of these uncertainties, I am very much for publication of the manuscript by Mnatsakanyan and co-workers. It contributes to the scientific debate on the nature of PTP and with very solid data supported by previous papers by Jonas group (and in most aspects by Bernardi group) argues that ATP synthase exhibits channel properties consistent with PTP activity in vivo.

We heartily thank the reviewers for all the insightful comments which led to the improvement of the manuscript.

We address each comment individually below.

Reviewer #1 (Remarks to the Author):

In the second round of revision, the authors have addressed the critiques from the previous round of review satisfactorily and the manuscript can now be recommended for publication in Nature Communications.

We thank the reviewer for the critical assessment of our manuscript and for all the suggestions and comments during this review process.

Reviewer #3 (Remarks to the Author):

In general, I find the results and the author's responses to questions convincing and satisfactory. However, when this work has been under revision the paper by Urbani A, et al. (Nat Commun. 10(1):4341. doi: 10.1038/s41467-019-12331-1) was published in which it was shown that the channel activity associated with highly purified bovine ATP synthase is observed when dimers and tetramers but NOT monomers of this protein complex are fused with planar lipid membranes. This work is crucial and should be cited and discussed in the current manuscript. In particular, it should be discussed why Mnatsakanyan could observe channel activity of the monomeric ATP synthase while all the data from Bernardi group and his co-workers indicate that PTP activity resides only in ATP synthase dimers.

In spite of these uncertainties, I am very much for publication of the manuscript by Mnatsakanyan and co-workers. It contributes to the scientific debate on the nature of PTP and with very solid data supported by previous papers by Jonas group (and in most aspects by Bernardi group) argues that ATP synthase exhibits channel properties consistent with PTP activity in vivo.

We thank the reviewer for raising so many important points during the review process, which significantly improved the manuscript. We now cite and discuss the key findings of Urbani et al. and the discrepancies between the two studies (please see Discussion).